# What does automatic differentiation compute for neural networks?

**Sejun Park**[1][*]   **Sanghyuk Chun**[2][*]   **Wonyeol Lee**[3]
[1]Korea University   [2]NAVER AI Lab   [3]Carnegie Mellon University

## Abstract

Forward- or reverse-mode automatic differentiation (AD) is a popular algorithm for computing the derivative of a function expressed by a program. AD always outputs the correct derivative if a program does not use any non-differentiable functions and control flows; however, it may return an arbitrary value otherwise. In this work, we investigate what AD computes for neural networks that may contain non-differentiable functions such as $\mathrm{ReLU}$ and maxpools. We first prove that AD always returns a generalized derivative called a Clarke subderivative for networks with pointwise activation functions, if the minibatch size is one and all non-differentiable neurons have distinct bias parameters. We show that the same conclusion does not hold otherwise, but does hold under some mild sufficient conditions. We also prove similar results for more general networks that can use maxpools and bias parameters shared across different neurons. We empirically check our sufficient conditions over popular network architectures and observe that AD almost always computes a Clarke subderivative in practical learning setups.

## 1 Introduction

Computing the derivative of a function represented by a program is a fundamental task in machine learning as well as many other areas such as scientific computing (Baydin et al., 2017; Heath, 2018). Automatic differentiation is a class of algorithms for this computation that are based on the chain rule, and has two popular "modes" called the forward mode and reverse mode (Griewank and Walther, 2008). In particular, the reverse mode includes the backpropagation algorithm (Rumelhart et al., 1986) as a special case and is implemented in diverse machine learning frameworks such as TensorFlow (Abadi et al., 2016), PyTorch (Paszke et al., 2017), and JAX (Frostig et al., 2018). This paper studies these two modes of automatic differentiation, and we write them simply as AD.

The correctness of AD has been extensively studied for decades, especially with respect to the *standard derivative*. For a program that consists of differentiable functions and no control flows (e.g., if-else and while statements), AD is shown to compute the standard derivative of the function represented by the program for *all* inputs (Elliott, 2018; Abadi and Plotkin, 2020; Brunel et al., 2020; Barthe et al., 2020; Huot et al., 2020; Vákár, 2021). If a program starts to use non-differentiable functions (e.g., $\mathrm{ReLU}$) or control flows, however, AD might not compute the standard derivative for some inputs (Kakade and Lee, 2018). Fortunately, even in this case, AD is shown to compute the standard derivative for *most* inputs under mild conditions which often hold in practice. For instance, Bolte and Pauwels (2020b); Lee et al. (2020); Mazza and Pagani (2021); Huot et al. (2023) proved that for all practically-used programs, AD does not compute the standard derivative at most on a measure-zero (i.e., negligible) subset of all real-valued inputs. In addition, Lee et al. (2023) studied the density of such inputs over all machine-representable (e.g., floating-point) inputs, proving that it is close to zero for many neural networks while it can be close to one if a network uses too many non-differentiable functions.

These correctness results show that AD computes the standard derivative at most inputs, yet often provide little information about what it computes at the remaining inputs. To better understand AD, several works have studied its correctness with respect to various notions of *generalized derivatives*, including the so-called Clarke subdifferential (Clarke, 1975). The Clarke subdifferential is one of the most traditional, widely-used generalized derivatives, which extends the subgradients of convex functions to non-convex functions and has been considered in several areas such as optimization

---

[*]Equal contribution
 Emails: {sejun.park000, sanghyuk.chun, wonyeol.lee.cs}@gmail.com

Table 1: A summary of correctness results of AD, with respect to the Clarke subdifferential. We put ✓ in the last column if the output of AD is always in the Clarke subdifferential, and ✗ otherwise.

| Reference | Distinct bias params | Shared bias params | Minibatch of size one | Choice of proxy derivatives | Without maxpools | Always correct? |
|---|---|---|---|---|---|---|
| Lee et al. (2023) | ✓ | | ✓ | ✓* | ✓ | ✓ |
| Ours (Theorem 1) | ✓ | | ✓ | ✓† | ✓ | ✓ |
| Ours (Lemma 2) | | | ✓ | ✓† | ✓ | ✗¶ |
| Ours (Lemma 3) | ✓ | | | ✓† | ✓ | ✗¶ |
| Ours (Theorem 4) | ✓ | | | ✓‡ | ✓ | ✓ |
| Ours (Theorem 6) | | ✓ | | ✓‡ | ✓ | ✓ |
| Ours (Lemma 7) | | ✓ | | ✓‡ | | ✗§ |

*AD uses either $D^-\rho(x)$ or $D^+\rho(x)$ as the proxy derivative of a pointwise activation function $\rho$ at $x$, where $D^-\rho$ and $D^+\rho$ denote the left-hand and right-hand derivatives of $\rho$.
†AD uses an element of the Clarke subdifferential as the proxy derivative of $\rho$.
‡AD uses $\lambda D^-\rho(x) + (1-\lambda)D^+\rho(x)$ as the proxy derivative of $\rho$ at $x$; $\lambda \in [0,1]$ is shared in a layer.
¶A sufficient condition for AD to be correct at the current parameter/input values is given in Theorem 5.
§A sufficient condition for AD to be correct at the current parameter/input values is given in Theorem 8.

and control theory (Clarke, 1990; Clarke et al., 1998). Some classical results such as (Clarke, 1990, Chapter 2) and (Rockafellar and Wets, 1998, Chapter 10) showed that the Clarke subdifferential enjoys the (exact) chain rule for certain classes of functions, implying that AD always computes an element of the Clarke subdifferential when applied to a program consisting of these functions. More recently, Lee et al. (2023) directly showed that AD always computes an element of the Clarke subdifferential when applied to a certain class of neural networks.

The previous results on AD and the Clarke subdifferential, however, are not applicable to many neural networks used in practice, e.g., networks that use non-differentiable functions (e.g., ReLU and maxpools) together with convolution layers, some normalization layers (e.g., BatchNorm), residual connections, or minibatches of inputs (Lee et al., 2023; Davis et al., 2020). This limitation makes it still unclear what AD computes for practical neural networks especially with respect to the Clarke subdifferential. We remark that some recent works such as (Bolte and Pauwels, 2020a;b; Lee et al., 2020; Huot et al., 2023) proved the correctness of AD over all inputs, with respect to fundamentally new notions of generalized derivatives (e.g., conservative or intensional derivatives); but they did not study how the output of AD is connected to the Clarke subdifferential, an arguably more popular notion of generalized derivatives.

**Contributions.** In this paper, we investigate what AD computes for neural networks in various problem setups. As in practice, we assume that for each non-differentiable function (e.g., ReLU), AD uses an element of its Clarke subdifferential as its "proxy derivative." In this setting, our first set of results is for neural networks with pointwise and piecewise-analytic activation functions (e.g., ReLU and HardSigmoid), which can be summarized as follows.

- Theorem 1 shows that AD *always* computes an element of the Clarke subdifferential, if the minibatch size is one and all non-differentiable neurons have distinct bias parameters. This generalizes the prior result in (Lee et al., 2023) which requires AD to use an element of the Bouligand subdifferential[1] for activation functions under a similar setup.

- Given this correctness result of AD with respect to the Clarke subdifferential, a natural question arises: does the same conclusion hold under non-trivial minibatch sizes or the absence of bias parameters? We prove this is not true. Lemmas 2 and 3 show that if the minibatch size is at least two or bias parameters are absent, then AD can return a value not in the Clarke subdifferential for some network, input, and parameter configuration.

- Then, without distinct bias parameters and the trivial minibatch size, when does AD compute an element of the Clarke subdifferential? Theorem 4 shows that for networks with distinct bias parameters, AD is always correct regardless of the minibatch size, as long as we choose proper

---

[1]The Bouligand subdifferential is a subset of the Clarke subdifferential. See Section 2.1 for their definitions.

proxy derivatives. For fully-connected networks that might not have distinct bias parameters, Theorem 5 provides an easily verifiable sufficient condition for checking the correctness of AD.

We next consider more general networks that can have shared bias parameters (e.g., as in convolutional layers) and maxpools (i.e., non-pointwise activation functions), under general minibatch sizes.

- Theorem 6 shows that for networks with shared bias parameters and no maxpools, AD always computes an element of the Clarke subdifferential as long as proper proxy derivatives are chosen. Namely, AD is correct with respect to the Clarke subdifferential for modern convolutional networks without maxpools. We also show that having shared bias parameters, no maxpools, and proper proxy derivatives are all necessary for this correctness result (Lemmas 2, 3, and 7).
- Theorem 8 provides a sufficient condition for verifying the correctness of AD when networks can have maxpools. Using our conditions in Theorems 5 and 8, we empirically check what AD outputs for fully-connected and convolutional networks under practical learning setups. In our experiments, we observe that AD successfully returns an element of the Clarke subdifferential although the non-differentiable points of activation functions are often touched during training.

**Organization.** In Section 2, we introduce notations and problem setup including the formal definitions of neural networks and AD. We present our main results on the correctness of AD for neural networks in Section 3. In Section 4, we empirically verify our sufficient conditions and check whether AD is correct under practical learning setups. We lastly conclude the paper in Section 5.

## 2 PROBLEM SETUP AND NOTATIONS

### 2.1 NOTATIONS

We first introduce the notations used in this paper. We use $\mathbb{N}$ and $\mathbb{R}$ to denote the set of positive integers and that of real numbers. For $n \in \mathbb{N}$, we use $\mathbf{1}_n \triangleq (1, \ldots, 1), \mathbf{0}_n \triangleq (0, \ldots, 0) \in \mathbb{R}^n$, $[n] \triangleq \{1, \ldots, n\}$, and we treat $x \in \mathbb{R}^n$ as a column vector. For $k, m_1, \ldots, m_k \in \mathbb{N}$ and a tensor $v \in \mathbb{R}^{m_1 \times \cdots \times m_k}$, we use $v_{i_1, \ldots, i_k}$ to denote the $(i_1, \ldots, i_k)$-th coordinate of $v$. Under the same setup, $v_{i_1, \ldots, i_j}$ for $j \in [k]$ denotes the $(k-j)$-dimensional tensor whose $(a_1, \ldots, a_{k-j})$-th coordinate is $v_{i_1, \ldots, i_j, a_1, \ldots, a_{k-j}}$. Likewise, for $n \in \mathbb{N}$, $f : \mathbb{R}^n \to \mathbb{R}^{m_1 \times \cdots \times m_k}$, and $j \in [k]$, we use $f_{i_1, \ldots, i_j} : \mathbb{R}^n \to \mathbb{R}^{m_{j+1} \times \cdots \times m_k}$ to denote the function such that $(f(x))_{i_1, \ldots, i_j} = f_{i_1, \ldots, i_j}(x)$ for all $x \in \mathbb{R}^n$; for $j = k$, we assume $m_{j+1} \times \cdots \times m_k = 1$ by following the convention. For $u = (u_1, \ldots, u_m)$ and $v = (v_1, \ldots, v_n)$, $u \oplus v \triangleq (u_1, \ldots, u_m, v_1, \ldots, v_n)$ denotes the concatenation of $u$ and $v$. For a matrix $A \in \mathbb{R}^{n \times m}$ whose $i$-th column is $a_i \in \mathbb{R}^n$, $\mathsf{vec}(A) \triangleq a_1 \oplus \cdots \oplus a_m$ denotes a vectorization of $A$. For $k, n_1, \ldots, n_k \in \mathbb{N}$, and given $x_{i_1, \ldots, i_k} \in \mathbb{R}$ for all $i_j \in [n_j]$ and $j \in [k]$, we write $[x_{i_1, \ldots, i_k}]_{i_1 \in [n_1], \ldots, i_k \in [n_k]}$ to denote the $k$-dimensional tensor whose $(i_1, \ldots, i_k)$-th coordinate is $x_{i_1, \ldots, i_k}$; we often use $[x_{i_1, \ldots, i_k}]_{i_1, \ldots, i_k}$ to denote this tensor when the range of $i_1, \ldots, i_k$ is clear from the context. We write $\mu_n$ to denote the $n$-dimensional Lebesgue measure. We often use signs $+$ and $-$ to denote $+1$ and $-1$, respectively; for example, for $s = -$ and $x \in \mathbb{R}$, $s \cdot x$ is $-x$.

For $f : \mathbb{R}^n \to \mathbb{R}^m$ and $x \in \mathbb{R}^n$ at which $f$ is differentiable, we use $Df(x) \in \mathbb{R}^{m \times n}$ to denote the Jacobian matrix of $f$ at $x$, and use $\mathsf{ndf}(f) \subset \mathbb{R}^n$ to denote the set of inputs at which $f$ is not differentiable. For $f : \mathbb{R} \to \mathbb{R}$ and $s \in \{-, +\}$, we use $D^s f(x) \triangleq \lim_{v \to x^s} Df(v)$. Given locally Lipschitz $f : \mathbb{R}^n \to \mathbb{R}$, the Bouligand subdifferential of $f$ at $x \in \mathbb{R}^n$ (Cui and Pang, 2021) is

$$\partial^{\mathsf{B}} f(x) \triangleq \left\{ s \in \mathbb{R}^n : \exists v_1, v_2, \ldots \in \mathbb{R}^n \setminus \mathsf{ndf}(f) \text{ such that } v_t \to x \text{ and } \nabla f(v_t) \to s \right\}.$$

The Clarke subdifferential of locally Lipschitz $f$ at $x \in \mathbb{R}^n$ (Clarke, 1990) is defined as the convex hull of $\partial^{\mathsf{B}} f(x)$, which we denote by $\partial^{\mathsf{C}} f(x)$. By definition, we always have $\partial^{\mathsf{B}} f(x) \subset \partial^{\mathsf{C}} f(x)$.

### 2.2 NEURAL NETWORKS

We define a neural network as follows. Given the number of layers $L \in \mathbb{N}$, let $N_0 \in \mathbb{N}$ be the dimension of input data, $M_l, N_l \in \mathbb{N}$ be the dimensions of intermediate vectors at layer $l \in [L]$, $W_l \in \mathbb{N}$ be the number of parameters at layer $l \in [L]$, and $W \triangleq W_1 + \cdots + W_L$. Further, given the minibatch size $B \in \mathbb{N}$, and for each $l \in [L]$, let $\tau_l : \mathbb{R}^{N_0 \times B} \times \cdots \times \mathbb{R}^{N_{l-1} \times B} \times \mathbb{R}^{W_l} \to \mathbb{R}^{M_l \times B}$ be an analytic *pre-activation function* and $\sigma_l : \mathbb{R}^{M_l \times B} \to \mathbb{R}^{N_l \times B}$ be a continuous *activation function*.

We also use $\ell : \mathbb{R}^{N_L \times B} \to \mathbb{R}$ to denote an analytic *loss function* that maps the last layer's output to a scalar-valued loss. In this setup, we consider a neural network as a function of model parameters. Specifically, given an input minibatch $X \in \mathbb{R}^{N_0 \times B}$ of size $B$, a *neural network* $\Psi(\,\cdot\,; X) : \mathbb{R}^W \to \mathbb{R}$ is defined recursively as follows: $z_0(w; X) \triangleq X$, and for all $l \in [L]$,

$$\Psi(w; X) \triangleq \ell(z_L(w; X)), \quad z_l(w; X) \triangleq \sigma_l(y_l(w; X)), \quad y_l(w; X) \triangleq \tau_l(z_{0:l-1}(w; X), w_l), \quad (1)$$

where $z_{i:j}(w; X) \triangleq (z_i(w; X), \ldots, z_j(w; X))$, $w \triangleq w_1 \oplus \cdots \oplus w_L$, and $w_l \triangleq (w_{l,1}, \ldots, w_{l,W_l}) \in \mathbb{R}^{W_l}$. Since the input minibatch $X$ is fixed while we compute the derivative of $\Psi$ with respect to $w$, we often omit $X$ and use $\Psi(w)$, $z_l(w)$, and $y_l(w)$ to denote $\Psi(w; X)$, $z_l(w; X)$, and $y_l(w; X)$.

We mainly focus on two classes of neural networks, where the first class is characterized as follows.

**Definition 1.** *A function $f : \mathbb{R} \to \mathbb{R}$ is "piecewise-analytic" if $f$ is continuous and there exist $n \in \mathbb{N}$, a partition $\{\mathcal{A}_i\}_{i \in [n]}$ of $\mathbb{R}$ consisting of intervals with $\mu_1(\mathcal{A}_i) > 0$ for all $i$, and analytic functions $\{f_i : \mathbb{R} \to \mathbb{R}\}_{i \in [n]}$ such that $f = f_i$ on $\mathcal{A}_i$ for all $i \in [n]$.*

**Condition 1.** *A neural network $\Psi$ satisfies $M_l = N_l$ for all $l \in [L]$, and*

$$\tau_l(x, w_l) = f_l(x, u_l) \text{ or } f_l(x, u_l) + b_l \mathbf{1}_B^\top, \qquad \sigma_l([x_{i,j}]_{i,j}) = [\rho_l(x_{i,j})]_{i,j},$$

*where $w_l = u_l$ or $w_l = u_l \oplus b_l$ for some $b_l \in \mathbb{R}^{N_l}$, $f_l$ is an analytic function, and $\rho_l : \mathbb{R} \to \mathbb{R}$ is a piecewise-analytic function for all $l \in [L]$. We say the network $\Psi$ has "distinct bias parameters" if $\tau_l(x, w_l) = f_l(x, u_l) + b_l \mathbf{1}_B^\top$ for all $l \in [L]$ with non-differentiable $\rho_l$.*

Here, $b_l$ denotes a vector of $N_l$ bias parameters, where each $b_{l,i}$ is used in computing the $i$-th row of $\tau_l$'s output (i.e., the $i$-th neuron). The neural networks satisfying Condition 1 cover a wide range of practical network architectures. For example, if $f_l(z_{0:l-1}, u_l) = A_l z_{l-1}$ where $u_l$ is a vectorization of some $A_l \in \mathbb{R}^{N_l \times N_{l-1}}$, then it represents a fully-connected layer. Likewise, $f_l$ in Condition 1 can represent attention layers (Vaswani et al., 2017), normalization layers (e.g., one-dimensional BatchNorm, and LayerNorm) (Ioffe and Szegedy, 2015; Ba et al., 2016), and their compositions. It can also express residual connections (He et al., 2016) as $f_l$ takes all previous activation tensors $z_{1:l-1}$. In addition, Condition 1 allows any pointwise and piecewise-analytic activation functions such as ReLU and HardSigmoid. Thus, Condition 1 covers not only simple neural networks such as fully-connected ones but also complex network architectures such as transformer-based ones.

Although Condition 1 can express a large class of practical networks, it cannot represent modern convolutional neural networks. For example, convolutional networks often have a single bias parameter that is shared across multiple neurons in the same channel, whereas Condition 1 does not allow such "shared" bias parameters. In addition, Condition 1 does not allow non-pointwise activation functions such as MaxPool2d. To cover these exceptional cases, we introduce Condition 2.

**Definition 2.** *A function $f : \mathbb{R}^n \to \mathbb{R}^m$ is a "maxpool" if there exist $\mathcal{I}_1, \ldots, \mathcal{I}_m \subset [n]$ such that*

$$f(x_1, \ldots, x_n) = \left( \max_{i \in \mathcal{I}_1} x_i, \ldots, \max_{i \in \mathcal{I}_m} x_i \right).$$

**Condition 2.** *A neural network $\Psi$ satisfies*

$$\tau_l(x, w_l) = f_l(x, u_l) \text{ or } f_l(x, u_l) + \sum_{c=1}^{C_l} b_{l,c} A_{l,c}, \qquad \sigma_l([x_{i,j}]_{i,j}) = \phi_l([\rho_l(x_{i,j})]_{i,j}),$$

*where $C_l \in \mathbb{N}$, $w_l = u_l$ or $w_l = u_l \oplus b_l$ for some $b_l = (b_{l,1}, \ldots, b_{l,C_l}) \in \mathbb{R}^{C_l}$, $A_{l,c} \in \{0,1\}^{M_l \times B}$ for all $c \in [C_l]$, $f_l$ is an analytic function, $\rho_l : \mathbb{R} \to \mathbb{R}$ is a piecewise-analytic function, and $\phi_l : \mathbb{R}^{M_l \times B} \to \mathbb{R}^{N_l \times B}$ is a maxpool for all $l \in [L]$. We say the network $\Psi$ has "shared bias parameters" if $\sum_{c=1}^{C_l} A_{l,c} = \mathbf{1}_{M_l} \mathbf{1}_B^\top$ for all $l \in [L]$ with non-differentiable $\rho_l$. We say the network $\Psi$ "has only trivial maxpools" if $\phi_l$ is an identity map for all $l \in [L]$.*

Condition 2 allows shared bias parameters (which are used in, e.g., convolutional layers and two-dimensional BatchNorm): $b_l$ denotes a vector of $C_l$ bias parameters, where each $b_{l,c}$ is used in computing the $(i, k)$-th output of $\tau_l$ (i.e., the $i$-th neuron for the $k$-th input) whenever $(A_{l,c})_{i,k} = 1$. Further, Condition 2 also allows non-pointwise activation functions (e.g., a composition of ReLU and MaxPool2d). Hence, it covers modern convolutional neural networks which often use normalization layers, maxpools, and residual connections. We note that Condition 2 with only trivial maxpools is a generalization of Condition 1: consider the case that $M_l = N_l = C_l$, $A_{l,c} = e_c \mathbf{1}_B^\top$, and $\phi_l$ is an identity map for all $l \in [L]$ and $c \in [C_l]$, where $e_c$ denotes the $c$-th standard basis of $\mathbb{R}^{C_l}$. Also, having shared bias parameters is a generalization of having distinct bias parameters.

## 2.3 AUTOMATIC DIFFERENTIATION

Automatic differentiation is a class of algorithms for computing the derivative of a function (represented by a program) based on the chain rule, and it has two popular modes: forward mode and reverse mode. In this paper, AD refers to these two modes of automatic differentiation. Given a neural network $\Psi$ defined by Eq. (1) and an input minibatch $X \in \mathbb{R}^{N_0 \times B}$, AD essentially computes

$$D^{\text{AD}}\Psi(\cdot\,;X) : \mathbb{R}^W \to \mathbb{R}^W$$

by applying the chain rule of differentiation to Eq. (1). In other words, for neural networks satisfying Condition 1 (or Condition 2), $D^{\text{AD}}\Psi$ is defined as the product of $D\tau_l$ and $D^{\text{AD}}\rho_l$ (and $D^{\text{AD}}\max_n$), where $\max_n$ denotes the max function over $\mathbb{R}^n$, and $D^{\text{AD}}\rho_l$ and $D^{\text{AD}}\max_n$ denote the "proxy gradients" of $\rho_l$ and $\max_n$ that AD uses in its computation. Here, the proxy gradients are necessary since $\rho_l$ and $\max_n$ are not differentiable in general, i.e., their (standard) derivatives might not exist at some points. We assume $D^{\text{AD}}\rho_l(x) \in \partial^{\text{C}}\rho_l(x)$ for all $x \in \mathbb{R}$, which implies $D^{\text{AD}}\rho_l(x) = D\rho_l(x)$ for all $x \notin \text{ndf}(\rho_l)$ since $\rho_l$ is piecewise-analytic (see Lemma 9). We also assume $D^{\text{AD}}\max_n(x) \triangleq e_i$ for all $x \in \mathbb{R}^n$, where $i$ depends on $x$ such that $\max_n(x) = x_i$, and $e_i$ denotes the $i$-th standard basis of $\mathbb{R}^n$.[2] Many AD systems, including TensorFlow and PyTorch, fulfill these two assumptions for all one-dimensional piecewise-analytic functions and all maxpool functions that are implemented in the systems. The formal expression of $D^{\text{AD}}\Psi$ for neural networks $\Psi$ can be found in Appendix B.

Throughout the paper, we say AD is "correct with respect to the Clarke subdifferential" (or simply, "correct") for a network $\Psi$, an input $X$, and parameters $w$ if $D^{\text{AD}}\Psi(w;X) \in \partial^{\text{C}}\Psi(w;X)$. For a fixed $\Psi$, we say AD is "always correct" if it is correct for all $X$ and $w$.

## 3 MAIN RESULTS

We are now ready to present our main results on the correctness of AD for neural networks, which consider various setups: the presence of bias parameters, the choice of the minibatch size, the choice of the proxy gradients used by AD, and the presence of maxpools. We first introduce our analyses on neural networks satisfying Condition 1 in Section 3.1. We then move to neural networks satisfying Condition 2 in Section 3.2. The proofs of all results in this section are given in Appendices D and E, and discussions on our theoretical results are provided in Sections H–J.

### 3.1 CORRECTNESS OF AD FOR NEURAL NETWORKS SATISFYING CONDITION 1

Our first result is about the correctness of AD for neural networks satisfying Condition 1, especially when there are distinct bias parameters and the minibatch size is one.

**Theorem 1.** *Let $\gamma \in \{\text{B}, \text{C}\}$ and $\Psi$ be a network satisfying Condition 1 with distinct bias parameters. Suppose that $D^{\text{AD}}\rho_l(x) \in \partial^\gamma \rho_l(x)$ for all $l \in [L]$ and $x \in \text{ndf}(\rho_l)$. Then, $D^{\text{AD}}\Psi(w;X) \in \partial^\gamma \Psi(w;X)$ for all $w \in \mathbb{R}^W$ and $X \in \mathbb{R}^{N_0 \times B}$ with $B = 1$.*

Theorem 1 states that if a network $\Psi$ has distinct bias parameters and the minibatch size is one, then AD computes an element of the Clarke (or Bouligand) subdifferential of $\Psi$ as long as the proxy gradient $D^{\text{AD}}\rho_l(x)$ is an element of the Clarke (or Bouligand) subdifferential for all $l$ and $x$. In other words, AD is *always correct* in this case. This result extends the previous correctness result in (Lee et al., 2023), which states that AD computes an element of the Clarke subdifferential under a stronger setup: $D^{\text{AD}}\rho_l(x) \in \partial^{\text{B}}\rho_l(x)$ and $\Psi$ has no residual connections; our result considers more general proxy gradients $D^{\text{AD}}\rho_l(x) \in \partial^{\text{C}}\rho_l(x)$ and allows residual connections.

To prove Theorem 1 when $\gamma = \text{B}$, we explicitly find a sequence $\eta_1, \eta_2, \ldots$ of parameters that converges to $w$ such that $\Psi$ is differentiable on the sequence and $D\Psi(\eta_1), D\Psi(\eta_2), \ldots$ converges to $D^{\text{AD}}\Psi(w)$, i.e., $D^{\text{AD}}\Psi(w) \in \partial^{\text{B}}\Psi(w)$. To construct such a sequence, we utilize bias parameters. First, observe that $D^{\text{AD}}\rho_l(x) \in \partial^{\text{B}}\rho_l(x)$ implies $D^{\text{AD}}\rho_l(y_{l,i}(w)) = D^{s_{l,i}}\rho_l(y_{l,i}(w))$ for some $s_{l,i} \in \{-, +\}$ for all $l \in [L]$ and $i \in [N_l]$ since $B = 1$; here, we choose $s_{l,i} = s_{l,i'}$ if $y_{l,i}(w) = y_{l,i'}(w)$. Then, we can find a sequence of bias parameters that converges to $b_{l,i}$ from the left/right side depending on $s_{l,i}$, while fixing non-bias parameters (i.e., $u_l$); this leads us to the statement of Theorem 1. Using the result for $\gamma = \text{B}$, we can also prove the case for $\gamma = \text{C}$. We note that

---

[2]This assumption implies $D^{\text{AD}}\max_n(x) \in \partial^{\text{B}}\max_n(x)$ for all $x \in \mathbb{R}^n$.

similar strategies are used to prove other correctness results in this paper that consider networks with (distinct/shared) bias parameters (Theorems 4 and 6). See Section D.1 for the detailed proof.

Given the correctness of AD under the presence of distinct bias parameters and $B = 1$, a natural question arises: are both conditions necessary? The following lemmas answer this by showing that both are indeed necessary for AD to be always correct; these lemmas are based on the incompatibility of the Clarke subdifferential with addition (Clarke et al., 1998; Kakade and Lee, 2018).

**Lemma 2.** *There exists a network $\Psi$ satisfying Condition 1 without distinct bias parameters such that $D^{\text{AD}}\rho_l(x) \in \partial^{\text{B}}\rho_l(x)$ for all $l \in [L]$ and $x \in \mathsf{ndf}(\rho_l)$, but $D^{\text{AD}}\Psi(w; X) \notin \partial^{\text{C}}\Psi(w; X)$ for some $w \in \mathbb{R}^W$ and $X \in \mathbb{R}^{N_0 \times B}$ with $B = 1$.*

**Lemma 3.** *There exists a network $\Psi$ satisfying Condition 1 with distinct bias parameters such that $D^{\text{AD}}\rho_l(x) \in \partial^{\text{B}}\rho_l(x)$ for all $l \in [L]$ and $x \in \mathsf{ndf}(\rho_l)$, but $D^{\text{AD}}\Psi(w; X) \notin \partial^{\text{C}}\Psi(w; X)$ for some $w \in \mathbb{R}^W$ and $X \in \mathbb{R}^{N_0 \times B}$ with $B \geq 2$.*

Lemmas 2 and 3 show the existence of (i) a network $\Psi$ satisfying Condition 1 without distinct bias parameters (respectively, with distinct bias parameters), (ii) an input minibatch $X$ of size $B = 1$ (respectively, of size $B \geq 2$), and (iii) a parameter configuration $w$, for which AD does not return an element of the Clarke subdifferential. In other words, AD can be incorrect in general if $B \geq 2$ or distinct bias parameters are absent.

Given these negative results, we may ask the following: is it impossible to have correct AD under more general setups? Our next result shows that if a network has distinct bias parameters, a proper choice of proxy gradients is sufficient for AD to be always correct, regardless of the minibatch size.

**Theorem 4.** *Let $\Psi$ be a network satisfying Condition 1 with distinct bias parameters. Suppose that there exist $\lambda_1, \ldots, \lambda_L \in [0, 1]$ such that $D^{\text{AD}}\rho_l(x) = \lambda_l D^-\rho_l(x) + (1 - \lambda_l)D^+\rho_l(x)$ for all $l \in [L]$ and $x \in \mathsf{ndf}(\rho_l)$. Then, $D^{\text{AD}}\Psi(w; X) \in \partial^{\text{C}}\Psi(w; X)$ for all $w \in \mathbb{R}^W$, $B \in \mathbb{N}$, and $X \in \mathbb{R}^{N_0 \times B}$. Further, if $\lambda_1, \ldots, \lambda_L \in \{0, 1\}$, then $D^{\text{AD}}\Psi(w; X) \in \partial^{\text{B}}\Psi(w; X)$ for all $w$, $B$, and $X$.*

Here, $D^-$ and $D^+$ denote the left- and right-hand derivatives (see Section 2.1 for formal definitions). Theorem 4 states that if the proxy gradient $D^{\text{AD}}\rho_l(x)$ is a convex combination of the left- and right-hand derivatives of $\rho_l$ with the same weight $\lambda_l$ for all $x \in \mathsf{ndf}(\rho_l)$, then AD is always correct for networks satisfying Condition 1 with distinct bias parameters, regardless of the minibatch size. For example, if $\rho_l$ has a single non-differentiable point $z \in \mathbb{R}$ (e.g., $z = 0$ for ReLU and LeakyReLU), then choosing $D^{\text{AD}}\rho_l(z) \in \partial^{\text{C}}\rho_l(z)$ is sufficient for satisfying this condition. If $|\mathsf{ndf}(\rho_l)| \geq 2$ as in ReLU6 and HardSigmoid,[3] then choosing $D^{\text{AD}}\rho_l(x) = D^-\rho_l(x)$ for all $x \in \mathsf{ndf}(\rho_l)$ (similarly, $D^{\text{AD}}\rho_l(x) = D^+\rho_l(x)$ for all $x \in \mathsf{ndf}(\rho_l)$) is sufficient to ensure the condition; we note that our counterexample in Lemma 3 does not satisfy this condition (see Section D.3). Therefore, the condition in Theorem 4 can be easily fulfilled by choosing proper proxy gradients of AD.

We next provide a sufficient condition for the correctness of AD under a general minibatch size and the (possible) absence of distinct bias parameters. For now, we focus on fully-connected networks; a result for general networks can be found in Theorem 8 in Section 3.2.

**Theorem 5.** *Let $\gamma \in \{\text{B}, \text{C}\}$ and $\Psi$ be a neural network satisfying Condition 1. Suppose that $\tau_l(z_{0:l-1}, w_l) = A_l z_{l-1}$ or $A_l z_{l-1} + b_l \mathbf{1}_B^\top$ for some $A_l \in \mathbb{R}^{N_l \times N_{l-1}}$ with $w_l = \mathsf{vec}(A_l)$ or $w_l = \mathsf{vec}(A_l) \oplus b_l$, and $D^{\text{AD}}\rho_l(x) \in \partial^\gamma\rho_l(x)$ for all $l \in [L]$ and $x \in \mathsf{ndf}(\rho_l)$. Then, for any $w \in \mathbb{R}^W$, $B \in \mathbb{N}$, and $X \in \mathbb{R}^{N_0 \times B}$, we have $D^{\text{AD}}\Psi(w; X) \in \partial^\gamma\Psi(w; X)$ if the columns of $z_{l-1}(w; X)$ are linearly independent whenever $y_{l,i,b}(w; X) \in \mathsf{ndf}(\rho_l)$ for some $l \in [L]$, $i \in [N_l]$, and $b \in [B]$.*

For fully-connected networks, Theorem 5 shows that the correctness of AD is guaranteed if the column vectors of $z_{l-1}(w; X)$ are linearly independent whenever $y_l(w; X)$ touches any non-differentiable point of the activation function $\rho_l$. If the input and hidden dimensions $N_0, \ldots, N_{L-1}$ are larger than the minibatch size $B$ (which often occurs in practical learning setups), this condition can be easily satisfied; we empirically demonstrate this in Section 4.

Unlike Theorems 1 and 4, a fully-connected network considered in Theorem 5 might not have distinct bias parameters. Hence, to prove the theorem for $\gamma = \text{B}$, we construct a sequence $\eta_1, \eta_2, \ldots$ of parameters that converges to $w$, by varying only the non-bias parameters (i.e., $A_l$) so that $D\Psi(\eta_1), D\Psi(\eta_2), \ldots$ converges to $D^{\text{AD}}\Psi(w)$. Here, the linear independence of the columns in $z_{l-1}(w; X)$ guarantees the existence of such a sequence. See Section D.5 for more details.

---

[3]ReLU6$(x) \triangleq \min\{\max\{0, x\}, 6\}$ and HardSigmoid $\triangleq \min\{\max\{0, x/6 + 1/2\}, 1\}$.

---

**Algorithm 1** Construction of $\mathcal{P}_l$

---

1: **Input:** $l$: the index of the target layer, $z_{0:l-1}$: the outputs of the $0, \ldots, (l-1)$-th layers, $w_{1:l}$: the parameters of the $1, \ldots, l$-th layers, $\{\mathcal{I}_j\}_{j \in [N_l] \times [B]}$: the index sets used by the maxpool function $\phi_l$ at the $l$-th layer.
2: **Initialize:** $\mathcal{P}_l \leftarrow \mathbb{R}^{W_l}$
3: **for** $i \in [M_l] \times [B]$ **do**
4:     $y_i \leftarrow \tau_{l,i}(z_{0:l-1}, w_l), \alpha_i \leftarrow \rho_l(y_i), \pi_i \leftarrow \partial y_i / \partial w_l$
5:     **if** $y_i \in \mathsf{ndf}(\rho_l)$ **then**
6:         Choose $s_i \in \{-, +\}$ such that $D^{\mathtt{AD}} \rho_l(y_i) = D^{s_i} \rho_l(y_i)$
7:         $\mathcal{P}_l \leftarrow \mathcal{P}_l \cap \{x \in \mathbb{R}^{W_l} : s_i \cdot \langle \pi_i, x \rangle > 0\}$
8:     **end if**
9: **end for**
10: **for** $j \in [N_l] \times [B]$ **do**
11:     $\mathcal{S}_j \leftarrow \operatorname{argmax}_{i \in \mathcal{I}_j} \alpha_i$
12:     **if** $|\mathcal{S}_j| \geq 2$ **then**
13:         $i^* \leftarrow$ the index in $\mathcal{S}_j$ that AD uses when computing $D^{\mathtt{AD}} \max(\{\alpha_i\}_{i \in \mathcal{I}_j})$
14:         $\mathcal{T}_j \leftarrow \{i \in \mathcal{S}_j : \partial^{\mathtt{AD}} \alpha_{i^*} / \partial w_{1:l} = \partial^{\mathtt{AD}} \alpha_i / \partial w_{1:l}\}^4$
15:         $\mathcal{P}_l \leftarrow \mathcal{P}_l \cap \big\{ x \in \mathbb{R}^{W_l} : \langle D^{\mathtt{AD}} \rho_l(y_{i^*}) \cdot \pi_{i^*}, x \rangle > \langle D^{\mathtt{AD}} \rho_l(y_i) \cdot \pi_i, x \rangle \text{ for all } i \in \mathcal{S}_j \setminus \mathcal{T}_j \big\}$
16:     **end if**
17: **end for**
18: **return** $\mathcal{P}_l$

---

## 3.2 CORRECTNESS OF AD FOR NEURAL NETWORKS SATISFYING CONDITION 2

In the previous subsection, we focused on networks satisfying Condition 1, and studied when AD is (always) correct or not by varying various setups such as the presence of bias parameters, the size of a minibatch, and the choice of the proxy gradients used by AD. In this subsection, we analyze the correctness of AD for neural networks satisfying Condition 2 (e.g., convolutional neural networks), which is a generalization of Condition 1, under a general minibatch size $B \in \mathbb{N}$. To this end, we first introduce the following theorem for networks with shared bias parameters and trivial maxpools.

**Theorem 6.** *Let $\Psi$ be a network satisfying Condition 2 with shared bias parameters and only trivial maxpools. Suppose that there exist $\lambda_1, \ldots, \lambda_L \in [0, 1]$ such that $D^{\mathtt{AD}} \rho_l(x) = \lambda_l D^- \rho_l(x) + (1 - \lambda_l) D^+ \rho_l(x)$ for all $l \in [L]$ and $x \in \mathsf{ndf}(\rho_l)$. Then, $D^{\mathtt{AD}} \Psi(w; X) \in \partial^{\mathtt{C}} \Psi(w; X)$ for all $w \in \mathbb{R}^W$, $B \in \mathbb{N}$, and $X \in \mathbb{R}^{N_0 \times B}$. Further, if $\lambda_1, \ldots, \lambda_L \in \{0, 1\}$, then $D^{\mathtt{AD}} \Psi(w; X) \in \partial^{\mathtt{B}} \Psi(w; X)$ for all $w \in \mathbb{R}^W$, $B \in \mathbb{N}$, and $X \in \mathbb{R}^{N_0 \times B}$.*

Theorem 6 is a generalization of Theorem 4: it ensures the correctness of AD over all inputs and parameters (under a proper choice of $D^{\mathtt{AD}} \rho_l$), as long as a network satisfying Condition 2 has shared bias parameters and only trivial maxpools at all layers. Many modern convolutional neural networks without maxpools satisfy the conditions in Theorem 6: e.g., MobileNet V3 (Howard et al., 2019), RexNet (Han et al., 2021), and ConvNext (Liu et al., 2022). Namely, AD is always correct for those networks although they contain non-differentiable activation functions.

Since Condition 2 is a generalization of Condition 1, choosing a proper $D^{\mathtt{AD}} \rho_l$ and having bias parameters are necessary for Theorem 6 as we observed in Section 3.1 (see Lemmas 2 and 3). Given the necessity of these conditions, our next result examines the only remaining condition in Theorem 6, which is about having only trivial maxpools.

**Lemma 7.** *There exists a network $\Psi$ satisfying Condition 2 with shared bias parameters and non-trivial maxpools such that $D^{\mathtt{AD}} \rho_l(x) = D^- \rho_l(x)$ for all $l \in [L]$ and $x \in \mathsf{ndf}(\rho_l)$, but $D^{\mathtt{AD}} \Psi(w; X) \notin \partial^{\mathtt{C}} \Psi(w; X)$ for some $w \in \mathbb{R}^W$, $B \in \mathbb{N}$, and $X \in \mathbb{R}^{N_0 \times B}$.*

Lemma 7 shows that AD can be incorrect with non-trivial maxpools. However, practical networks, especially convolutional networks, often include them. To examine the correctness of AD for such networks in practice, we provide a generic sufficient condition that guarantees correct AD.

**Theorem 8.** *Let $\Psi$ be a network satisfying Condition 2. Suppose that $D^{\mathtt{AD}} \rho_l(x) \in \partial^{\mathtt{B}} \rho_l(x)$ for all $l \in [L]$ and $x \in \mathsf{ndf}(\rho_l)$. Then, for any $w \in \mathbb{R}^W$, $B \in \mathbb{N}$, and $X \in \mathbb{R}^{N_0 \times B}$, it holds that $D^{\mathtt{AD}} \Psi(w; X) \in \partial^{\mathtt{B}} \Psi(w; X)$ if $\mathcal{P}_l \neq \emptyset$ for all $l \in [L]$ where $\mathcal{P}_l$ denotes the output of Algorithm 1.*

---

$^4 \partial^{\mathtt{AD}} \alpha_i / \partial w_{1:l}$ denotes the partial derivative of $\alpha_i$ with respect to $w_{1:l}$ that reverse-mode AD computes.

To describe the main intuition behind $\mathcal{P}_l$, we first consider the case when each $\mathcal{T}_j$ in Algorithm 1 has only a single element $i^*$. In this case, $\mathcal{P}_l$ satisfies the following property: for any $\zeta_l \in \mathcal{P}_l$ and small enough $\varepsilon > 0$, the activation function $\sigma_l$ is differentiable at $\tau_l(z_{0:l-1}(w), w_l + \varepsilon\zeta_l)$ and

$$D\sigma_l\Big(\tau_l(z_{0:l-1}(w), w_l + \varepsilon\zeta_l)\Big) \to D^{\mathrm{AD}}\sigma_l\Big(\tau_l(z_{0:l-1}(w), w_l)\Big) \quad \text{as } \varepsilon \to 0, \tag{2}$$

where $D^{\mathrm{AD}}\sigma_l$ denotes the proxy derivative of $\sigma_l$ used by AD (see Appendix B for its formal definition). Due to this property, whenever $\mathcal{P}_l$ is not empty for all $l \in [L]$, we can construct a sequence of parameters for showing $D^{\mathrm{AD}}\Psi(w) \in \partial^{\mathrm{B}}\Psi(w)$ as in the proof of Theorem 1. To construct $\mathcal{P}_l$ satisfying Eq. (2), Algorithm 1 adds constraints to $\mathcal{P}_l$ in Lines 3–9 and 10–17, which ensure

$$D\rho_l\Big(\tau_{l,i,b}(z_{0:l-1}(w), w_l + \varepsilon\zeta_l)\Big) \to D^{\mathrm{AD}}\rho_l\Big(\tau_{l,i,b}(z_{0:l-1}(w), w_l)\Big) \quad \text{as } \varepsilon \to 0 \quad \text{and} \tag{3}$$

$$D\phi_l\Big(\rho_l(\tau_l(z_{0:l-1}(w), w_l + \varepsilon\zeta_l))\Big) \to D^{\mathrm{AD}}\phi_l\Big(\rho_l(\tau_l(z_{0:l-1}(w), w_l))\Big) \quad \text{as } \varepsilon \to 0 \tag{4}$$

for all $i \in [N_l]$ and $b \in [B]$; and Eqs. (3) and (4) immediately imply Eq. (2). Here, $D^{\mathrm{AD}}\phi_l$ denotes the proxy derivative of $\phi_l$ used by AD, which is induced by $D^{\mathrm{AD}}\max_n$ (defined in Section 2.3). Now, we consider the remaining case when $\mathcal{T}_j$ has at least two elements. In this case, Eq. (4) might not hold. Yet, even without Eq. (4), we can still find a sequence of parameters $\eta_1, \eta_2, \ldots \in \mathbb{R}^W$ such that $\eta_n \to w$ and $D\Psi(\eta_n) \to D^{\mathrm{AD}}\Psi(w)$ as $n \to \infty$, using the definition of $\mathcal{T}_j$ (Line 14): $\partial^{\mathrm{AD}}\alpha_{i^*}/\partial w_{1:l} = \partial^{\mathrm{AD}}\alpha_i/\partial w_{1:l}$ for all $i \in \mathcal{T}_j$. See Section E.3 for detailed arguments.

To verify the sufficient condition in Theorem 8, one needs to check whether $\mathcal{P}_l$ is empty or not, where $\mathcal{P}_l = \{x \in \mathbb{R}^{W_l} : \langle a_{l,i}, x \rangle > 0 \text{ for all } i \in [k_l]\}$ for some $a_{l,1}, \ldots, a_{l,k_l} \in \mathbb{R}^{W_l}$ (which are described in Algorithm 1). To check this, one can solve the following linear programming: find $(c, x) \in \mathbb{R} \times \mathbb{R}^{W_l}$ such that it maximizes $c$ subject to $\langle a_{l,i}, x \rangle \geq c$ for all $i \in [k_l]$. Then, the solution $c$ of this problem is strictly positive if and only if $\mathcal{P}_l$ is not empty.

## 4 EXPERIMENTS

We use the sufficient conditions in Theorems 5 and 8 to verify whether AD is correct in two practical learning scenarios. In the first scenario, we consider fully-connected networks with distinct bias parameters that do not satisfy the conditions in Theorems 1 and 4. In the second one, we use convolutional networks with shared bias parameters that do not satisfy the conditions in Theorem 6.

**Scenario 1: Fully-connected networks.** We consider three fully-connected networks with two hidden layers and hidden dimensions of $N_1 = 256$ and $N_2 = 64$, where each network uses one of ReLU6, HardTanh, and HardSigmoid as its non-differentiable activation function. We trained these networks on the MNIST dataset (LeCun et al., 2010) using stochastic gradient descent (SGD) with the minibatch size $B = 128$, where each gradient was computed via (reverse-mode) AD implemented in PyTorch. All networks were trained for 20 epochs with the initial learning rate 0.05 and the weight decay 0.0001, where the learning rate was decayed by the cosine annealing scheduling (Loshchilov and Hutter, 2017). We note that all the activation functions in these networks (i.e., $\rho_l \in \{\text{ReLU6}, \text{HardTanh}, \text{HardSigmoid}\}$) have exactly two non-differentiable points, and PyTorch uses the following as their proxy gradients: $D^{\mathrm{AD}}\rho_l(x) = 0$ for all $x \in \mathsf{ndf}(\rho_l)$. Due to this, there is no $\lambda_l \in [0,1]$ satisfying $D^{\mathrm{AD}}\rho_l(x) = \lambda_l D^-\rho_l(x) + (1-\lambda_l)D^+\rho_l(x)$ for all $x \in \mathsf{ndf}(\rho_l)$, i.e., these networks do not satisfy the conditions in Theorems 1 and 4.

During the training, we checked if AD is correct by verifying the sufficient condition in Theorem 5, i.e., whether the activation matrix $z_{l-1}(w) \in \mathbb{R}^{N_l \times B}$ has full column rank whenever any non-differentiable point of $\rho_l$ is touched. We ran five experiments for each network, and observed that all the parameter values $w \in \mathbb{R}^W$ taken over all training steps (469 steps/epoch $\times$ 20 epoch) satisfied the sufficient condition. This observation and Theorem 5 imply that in our experiments, AD returned an element of the Clarke subdifferential at all training steps, even though the non-differentiable points of activation functions were touched for the following number of times in each training (averaged over five runs): 0, 9.8, and 13.8 times for networks using ReLU6, HardTanh, and HardSigmoid, respectively.

**Scenario 2: Convolutional networks with maxpools.** We next evaluate the correctness of AD for three convolutional networks with non-trivial maxpools: VGG11 (Simonyan and Zisserman, 2015); VGG11 with BatchNorm (VGG11-BN), which adds the BatchNorm operation after each

Table 2: Statistics related to $\mathcal{S}_j$ and $\mathcal{S}_j \setminus \mathcal{T}_j$. The total number of $\max_n$ operations in VGG11 and VGG11-BN per each layer (with a non-trivial maxpool) is 1,048,576 / 524,288 / 262,144 / 131,072 / 32,768. ResNet has 2,095,140 $\max_n$ operations in the first layer. The total number of training steps is 7,820 for all networks. All the below values are averaged over five independent runs.

| Network | Ratio of $\max_n$ with $\lvert\mathcal{S}_j\rvert \geq 2$ (%) | # training steps with $\mathcal{S}_j \setminus \mathcal{T}_j \neq \emptyset$ | Average $\lvert\mathcal{S}_j \setminus \mathcal{T}_j\rvert$ when $\mathcal{S}_j \setminus \mathcal{T}_j \neq \emptyset$ | Correct? |
|---|---|---|---|---|
| VGG11 | 3.2 / 0.04 / $2.1{\times}10^{-6}$ / $1.7{\times}10^{-6}$ / 0 | 1435.8 / 166.3 / 44.0 / 18.0 / 0 | 1.18 / 1.04 / 1.00 / 1.00 / NA | ✓ |
| VGG11-BN | 4.1 / 0.07 / $4.5{\times}10^{-6}$ / $4.2{\times}10^{-6}$ / $3.5{\times}10^{-6}$ | 3668.5 / 453.0 / 91.0 / 42.0 / 9.0 | 1.46 / 1.09 / 1.01 / 1.02 / 1.00 | ✓ |
| ResNet18 | 4.2 | 3904.8 | 1.78 | ✓ |

convolution operation in VGG11; and ResNet18 (He et al., 2016). In VGG11 and VGG11-BN, five layers have non-trivial maxpools, and in ResNet18, only the first layer has a non-trivial maxpool. In these networks, ReLU is the only pointwise non-differentiable activation function, and PyTorch uses $D^{\mathrm{AD}}\mathrm{ReLU}(0) = 0$ as its proxy gradient at zero. To reduce the computational overhead, we halved the channel dimensions of VGG11 and VGG11-BN. We trained these networks on the CIFAR-10 dataset (Krizhevsky et al., 2009) using AD-based SGD with the same settings described above.

During the training, we checked if the parameter values $w$ at each training step satisfy the sufficient condition in Theorem 8. To construct the set $\mathcal{P}_l$ in Theorem 8, we used Algorithm 2 in Section F, which is identical to Algorithm 1 for networks that have shared bias parameters and use ReLU (with $D^{\mathrm{AD}}\mathrm{ReLU}(0) = 0$) as the only non-differentiable $\rho_l$. The major difference between Algorithms 1 and 2 is that Algorithm 2 adds constraints to $\mathcal{P}_l$ only when a tie occurs at some $\max_n$ operation in maxpools, and it does not care whether any input to ReLU touches zero or not. In these experiments, we observed that the sufficient condition in Theorem 8 was always satisfied, implying that AD always returned an element of the Clarke subdifferential.

To better understand the results, we measured three additional quantities. The first one is the number of events that a tie occurs at some $\max_n$, i.e., $\lvert\mathcal{S}_j\rvert \geq 2$ in Algorithm 2. We counted the number of such events over all $\max_n$ operations in each layer (with a non-trivial maxpool), and divided it by the total number of $\max_n$ operations in that layer, which we denote by "ratio of $\max_n$ with $\lvert\mathcal{S}_j\rvert \geq 2$." As summarized in Table 2, we observed that this ratio is not negligible (e.g., $\geq 3\%$ for the first layer). This is because, for some $\max_n$ operation, the image patches corresponding to the receptive fields of its inputs are often identical (see Section G for concrete examples); for such a $\max_n$ operation, a tie always occurs regardless of the parameter values $w$.

To exclude such trivial cases and examine non-trivial ties in $\max_n$ operations, we also measured the number of training steps where $\lvert\mathcal{S}_j \setminus \mathcal{T}_j\rvert \geq 1$, and the average size of $\mathcal{S}_j \setminus \mathcal{T}_j$ when $\lvert\mathcal{S}_j \setminus \mathcal{T}_j\rvert \geq 1$, both per each layer with a non-trivial maxpool. As shown in Table 2, the event that $\lvert\mathcal{S}_j \setminus \mathcal{T}_j\rvert \geq 1$ (i.e., non-trivial ties exist) occurred frequently during training: e.g., in VGG11, this event happened in the first layer at 1435.8 steps (on average) among the total of 7,820 training steps. We also observed that the size of $\mathcal{S}_j \setminus \mathcal{T}_j$ was typically one or two, implying that $\mathcal{P}_l$ was non-empty very easily.

## 5 CONCLUSION

In this paper, we study the correctness of AD for neural networks. We first show that AD is always correct for networks satisfying Condition 1 if they have distinct bias parameters and the minibatch size is one (Theorem 1). While AD may not be always correct if one of the conditions in Theorem 1 is violated (Lemmas 2 and 3), we prove that having proper proxy derivatives ensures AD to be correct again for general minibatch sizes (Theorem 4), under the presence of distinct bias parameters. For general fully-connected networks that may not have distinct bias parameters, we provide a sufficient condition for checking the correctness of AD (Theorem 5), which often holds in practical learning setups. We also prove similar results for a more general class of networks that can have shared bias parameters and maxpools (Theorem 6 and Lemma 7), and provide a generic sufficient condition as well (Theorem 8), which often holds in the training of practical convolutional networks (Section 4). We believe our results and analyses would contribute to a better understanding of AD.

ACKNOWLEDGEMENTS

SP was supported by Institute of Information & communications Technology Planning & Evaluation (IITP) grant funded by the Korea government (MSIT) (No. 2019-0-00079, Artificial Intelligence Graduate School Program, Korea University) and Basic Science Research Program through the National Research Foundation of Korea (NRF) funded by the Ministry of Education (2022R1F1A1076180).

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

## A   ADDITIONAL NOTATIONS AND DEFINITIONS

For $n, m \in \mathbb{N}$, we use $\mathbf{0}_{n \times m}$ (respectively, $\mathbf{1}_{n \times m}$) to denote the matrix consisting of zeros (respectively, ones) of size $n \times m$. For $x \in \mathbb{R}$, we define

$$\mathrm{sign}(x) \triangleq \begin{cases} +1 \ (\text{or } +) & \text{if } x > 0 \\ 0 & \text{if } x = 0 \\ -1 \ (\text{or } -) & \text{if } x < 0. \end{cases}$$

For $\mathcal{S} \subset \mathbb{R}$, we use $\mathrm{int}(\mathcal{S})$ to denote the interior of $\mathcal{S}$. For $\mathcal{S} \subset \mathbb{R}^n$ (or a set $\mathcal{S}$ of functions from $\mathbb{R}^n$ to $\mathbb{R}^m$), we use $\mathrm{conv}(\mathcal{S})$ to denote the convex hull of $\mathcal{S}$: $\mathrm{conv}(\mathcal{S}) \triangleq \{ \sum_{i \in [K]} c_i x_i : K \in \mathbb{N}, c_i \geq 0, x_i \in \mathcal{S} \text{ with } \sum_{i \in [K]} c_i = 1 \}$.

For a piecewise-analytic function $f : \mathbb{R} \to \mathbb{R}$ with a partition $\{\mathcal{A}_i\}_{i \in [n]}$ and $x \in \mathbb{R}$, we use $\mathcal{P}(f, x, +)$ to denote $\mathcal{A}_i$ for some $i \in [n]$ such that $(x, x + \varepsilon) \subset \mathcal{A}_i$ for some $\varepsilon > 0$. Likewise, we use $\mathcal{P}(f, x, -)$ to denote $\mathcal{A}_i$ for some $i \in [n]$ such that $(x - \varepsilon, x) \subset \mathcal{A}_i$ for some $\varepsilon > 0$. For a differentiable function $f : \mathbb{R}^{n_1} \times \cdots \times \mathbb{R}^{n_k} \to \mathbb{R}$ and $x_i \in \mathbb{R}^{n_i}$ for all $i \in [k]$, we use $D_i f(x_1, \ldots, x_k)$ to denote the derivative of $f$ with respect to the $i$-th argument in $(x_1, \ldots, x_k)$, i.e., $\partial f(x_1, \ldots, x_k)/\partial x_i \in \mathbb{R}^{n_i}$.

We say that a function $f : \mathbb{R}^n \to \mathbb{R}^m$ is a "differentiable maxpool" if it is a maxpool with $|\mathcal{I}_1| = \cdots = |\mathcal{I}_m| = 1$ in Definition 2, that is, $f(x_1, \ldots, x_n) = (x_{i_1}, \ldots, x_{i_m})$ for some $i_1, \ldots, i_m \in [n]$. We note that a trivial maxpool is always a differentiable maxpool.

## B   AUTOMATIC DIFFERENTIATION FOR NEURAL NETWORKS

Given a neural network $\Psi$, its parameters $w \in \mathbb{R}^W$, and an input minibatch $X \in \mathbb{R}^{N_0 \times B}$, the output of AD $D^{\mathrm{AD}} \Psi(w; X)$ can be written by the following recursive relationship that involves $D^{\mathrm{AD}} z_l(w; X) \in \mathbb{R}^{N_l \times B \times W}$ and $D^{\mathrm{AD}} y_l(w; X) \in \mathbb{R}^{M_l \times B \times W}$: for $l \in [L]$, $i \in [N_l]$, $j \in [M_l]$, and $b \in [B]$,

$$D^{\mathrm{AD}} \Psi(w; X) \triangleq \sum_{(i', b') \in [N_L] \times [B]} \big( D\ell\big(z_L(w; X)\big)\big)_{i', b'} \cdot \big(D^{\mathrm{AD}} z_L(w; X)\big)_{i', b'} \in \mathbb{R}^W,$$

$$\big(D^{\mathrm{AD}} z_l(w; X)\big)_{i, b} \triangleq \sum_{(j', b') \in [M_l] \times [B]} \big(D^{\mathrm{AD}} \sigma_l\big(y_l(w; X)\big)\big)_{i, b, j', b'} \cdot \big(D^{\mathrm{AD}} y_l(w; X)\big)_{j', b'} \in \mathbb{R}^W,$$

$$\big(D^{\mathrm{AD}} y_l(w; X)\big)_{j, b} \triangleq \sum_{k \in [l-1]} \sum_{(i', b') \in [N_k] \times [B]} \big(D_k \tau_{l, j, b}\big(z_0(w; X), \ldots, z_{l-1}(w; X), w_l\big)\big)_{i', b'} \cdot \big(D^{\mathrm{AD}} z_k(w; X)\big)_{i', b'}$$
$$+ \sum_{i' \in [W_l]} \big(D_l \tau_{l, j, b}\big(z_0(w; X), \ldots, z_{l-1}(w; X), w_l\big)\big)_{i'} \cdot e_{W_1 + \cdots + W_{l-1} + i'} \in \mathbb{R}^W,$$

where $e_i$ denotes the $i$-th standard basis of $\mathbb{R}^W$. Here, $D^{\mathrm{AD}} \sigma_l(x) \in \mathbb{R}^{N_l \times B \times M_l \times B}$ is defined as follows for all $x \in \mathbb{R}^{M_l \times B}$: when the network $\Psi$ satisfies Condition 1, we use

$$\big(D^{\mathrm{AD}} \sigma_l(x)\big)_{i, b, j, c} \triangleq \begin{cases} D^{\mathrm{AD}} \rho_l(x_{i, b}) & \text{if } (j, c) = (i, b) \\ 0 & \text{if } (j, c) \neq (i, b), \end{cases}$$

and when $\Psi$ satisfies Condition 2, we use

$$\big(D^{\mathrm{AD}} \sigma_l(x)\big)_{i, b, j, c} \triangleq \begin{cases} D^{\mathrm{AD}} \rho_l(x_{i, b}) & \text{if } (j, c) = (j_{i, b}^*, c_{i, b}^*) \\ 0 & \text{if } (j, c) \neq (j_{i, b}^*, c_{i, b}^*), \end{cases}$$

where $(j_{i, b}^*, c_{i, b}^*) \in \mathrm{argmax}_{(j', c') \in \mathcal{I}_{i, b}} x_{j', c'}$ denotes the index that AD uses for the proxy gradient of $(\phi_l)_{i, b} : \mathbb{R}^{M_l \times B} \to \mathbb{R}$ (i.e., the maxpool in the $l$-th layer restricted to its $(i, b)$-th output) and $\mathcal{I}_{i, b} \subset [M_l] \times [B]$ denotes the set of indices that $(\phi_l)_{i, b}$ uses (i.e., $\phi_l$ uses to compute its $(i, b)$-th output). We note that the above definition of $D^{\mathrm{AD}} \Psi(w; X)$ depends on the choice of proxy gradients $D^{\mathrm{AD}} \rho_l : \mathbb{R} \to \mathbb{R}$. Whenever we want to make this dependency explicit, we will write $D^{\mathrm{AD}} \Psi_{(D^{\mathrm{AD}} \rho_1, \ldots, D^{\mathrm{AD}} \rho_L)}(w; X)$ to denote $D^{\mathrm{AD}} \Psi(w; X)$.

## C   TECHNICAL LEMMAS FOR MAIN PROOFS

**Lemma 9.** *Let $f : \mathbb{R} \to \mathbb{R}$ be a piecewise-analytic function and $x \in \mathbb{R}$. Then, $f$ is differentiable at $x$ if and only if $f$ is continuously differentiable at $f$.*

*Proof of Lemma 9.* We only prove that if $f$ is differentiable at $x$, then $f$ is continuously differentiable at $x$, since the converse is trivial. Suppose that $f$ is differentiable at $x$. Then, by the definition of piecewise-analyticity (Definition 1), there exist analytic functions $f^-, f^+ : \mathbb{R} \to \mathbb{R}$ and $a < x < b$ such that $f = f^-$ on $(a, x)$ and $f = f^+$ on $(x, b)$. Since $f$ is differentiable at $x$ and $f^-, f^+$ are analytic over $\mathbb{R}$, we have

$$\lim_{z \to x^-} Df(z) = \lim_{z \to x^-} Df^-(z) = Df^-(x), \qquad \lim_{z \to x^+} Df(z) = \lim_{z \to x^+} Df^+(z) = Df^+(x),$$

$$Df^-(x) = \lim_{z \to x^-} \frac{f^-(z) - f^-(x)}{z - x} = Df(x) = \lim_{z \to x^+} \frac{f^+(z) - f^+(x)}{z - x} = Df^+(x).$$

Here, the second and third equalities in the second line above follow from $f$ being differentiable at $x$. This implies that $f$ is continuously differentiable at $x$. $\qquad\square$

**Lemma 10.** *For any open $\mathcal{O} \subset \mathbb{R}$, analytic and non-constant $f : \mathcal{O} \to \mathbb{R}$, and $x \in \mathcal{O}$, there exists $\varepsilon > 0$ such that*

$$f(x) \notin f([x - \varepsilon, x + \varepsilon] \setminus \{x\}).$$

*Furthermore, $f$ is strictly monotone on $[x, x + \varepsilon]$ and strictly monotone on $[x - \varepsilon, x]$. In particular, if $Df(x) > 0$ (or $Df(x) < 0$), then $f$ is strictly increasing (or strictly decreasing) on $[x - \varepsilon, x + \varepsilon]$.*

*Proof of Lemma 10.* Without loss of generality, suppose that $f(x) = 0$. Since $f$ is analytic, $f$ is infinitely differentiable and can be represented by the Taylor series on $(x - \delta, x + \delta)$ for some $\delta > 0$ as follows:

$$f(z) = \sum_{i=0}^{\infty} \frac{f^{(i)}(x)}{i!} (z - x)^i \quad \text{on} \quad (x - \delta, x + \delta)$$

where $f^{(i)}$ denotes the $i$-th derivative of $f$. Since $f$ is non-constant, there exists $i \in \mathbb{N}$ such that $f^{(i)}(x) \neq 0$. Let $i^*$ be the minimum such $i$. Then, by the Taylor's theorem,

$$f(z) = \frac{f^{(i^*)}(x)}{i^*!} (z - x)^{i^*} + o(|z - x|^{i^*}) \text{ on } (x - \delta, x + \delta). \tag{5}$$

We now consider the case that $f^{(i^*)}(x) > 0$ and $i^*$ is odd. By Eq. (5) and the analyticity of $f$, we can choose $\varepsilon \in (0, \delta)$ so that

(i)  $f(z) < 0$ on $[x - \varepsilon, x)$ and $f(z) > 0$ on $(x, x + \varepsilon]$, and

(ii)  $f^{(i^*)}(z) > 0$ on $[x - \varepsilon, x + \varepsilon]$.

Then, by *(i)*, $f(x) \notin f([x - \varepsilon, x + \varepsilon] \setminus \{x\})$. Furthermore, we claim that $f$ is strictly monotone on $[x - \varepsilon, x]$ and $[x, x + \varepsilon]$. By *(ii)*, the analyticity of $f$, the mean value theorem, and by our assumption that $f^{(i)}(x) = 0$ for all $i < i^*$, it holds that

$$f^{(i^*-1)}(z) < 0 \ \text{ on } \ (x - \varepsilon, x) \quad \text{and} \quad f^{(i^*-1)}(z) > 0 \ \text{ on } \ (x, x + \varepsilon).$$

Again, by the same reasoning, we have

$$f^{(i^*-2)}(z) > 0 \ \text{ on } \ (x - \varepsilon, x) \quad \text{and} \quad f^{(i^*-2)}(z) > 0 \ \text{ on } \ (x, x + \varepsilon).$$

By repeating this process, one can show that $f^{(1)}(z) > 0$ on $(x - \varepsilon, x)$ and $f^{(1)}(z) > 0$ on $(x, x + \varepsilon)$, implying that $f$ is strictly monotone on $[x - \varepsilon, x]$ and $[x, x + \varepsilon]$.

By applying similar arguments, one can also show that the above claim (i.e., $f(x) \notin f([x - \varepsilon, x + \varepsilon] \setminus \{x\})$ and $f$ is strictly monotone on $[x - \varepsilon, x]$ and $[x, x + \varepsilon]$ for some $\varepsilon > 0$) holds in the remaining cases: $f^{(i^*)}(x) < 0$ and $i^*$ is odd; $f^{(i^*)}(x) > 0$ and $i^*$ is even; and $f^{(i^*)}(x) < 0$ and $i^*$ is even. This completes the proof of Lemma 10. $\qquad\square$

**Lemma 11.** *Let $f_1, f_2 : \mathbb{R} \to \mathbb{R}$ be analytic functions, $\rho : \mathbb{R} \to \mathbb{R}$ be a piecewise-analytic function, and $z \in \mathbb{R}$. Let $D^{\text{AD}}\rho(x) \in \partial^{\text{B}}\rho(x)$ for all $x \in \text{ndf}(\rho)$. Suppose that*

*(i) for each $i \in \{1, 2\}$, if $f_i(z) \in \text{ndf}(\rho)$, then $D^{\text{AD}}\rho(f_i(z)) = D^{s_i}\rho(f_i(z))$ for some $s_i \in \{-, +\}$ and $s_i \cdot Df_i(z) > 0$ holds; and*

*(ii) $\rho(f_1(z)) \geq \rho(f_2(z))$ and $D^{\text{AD}}\rho(f_1(z)) \cdot Df_1(z) > D^{\text{AD}}\rho(f_2(z)) \cdot Df_2(z)$.*

*Then, it holds that*
$$\lim_{\varepsilon \to 0^+} \text{sign}\big(\rho(f_1(z + \varepsilon)) - \rho(f_2(z + \varepsilon))\big) = +.$$

*Proof of Lemma 11.* If $f_i(z) \in \text{ndf}(\rho)$, then $\text{sign}(Df_i(z)) = s_i \in \{-, +\}$ by *(i)*, so $f_i$ is non-constant. Therefore, by Lemma 10, we have
$$\lim_{\varepsilon \to 0^+} \text{sign}\big(f_i(z + \varepsilon) - f_i(z)\big) = s_i, \tag{6}$$
and this implies that
$$D^{\text{AD}}\rho(f_i(z)) \cdot Df_i(z) = D^{s_i}\rho(f_i(z)) \cdot Df_i(z) = D^+(\rho \circ f_i)(z),$$
where the first equality is by *(i)* and the second equality is by the piecewise-analyticity of $\rho$, the chain rule, and Eq. (6). If $f_i(z) \notin \text{ndf}(\rho)$, then it is easy to observe that
$$D^{\text{AD}}\rho(f_i(z)) \cdot Df_i(z) = D\rho(f_i(z)) \cdot Df_i(z) = D(\rho \circ f_i)(z) = D^+(\rho \circ f_i)(z),$$
where the first equality is by the assumption that $D^{\text{AD}}\rho(x) \in \partial^{\text{B}}\rho(x)$, and the second equality is by the chain rule. Hence, by *(ii)*, we have
$$D^{\text{AD}}\rho(f_1(z)) \cdot Df_1(z) - D^{\text{AD}}\rho(f_2(z)) \cdot Df_2(z) = D^+(\rho \circ f_1 - \rho \circ f_2)(z) > 0.$$
This implies that for any small enough $\varepsilon > 0$,
$$\begin{aligned}
+ = \text{sign} &\left( \frac{\rho(f_1(z + \varepsilon)) - \rho(f_1(z))}{\varepsilon} - \frac{\rho(f_2(z + \varepsilon)) - \rho(f_2(z))}{\varepsilon} \right) \\
&= \text{sign}\big(\rho(f_1(z + \varepsilon)) - \rho(f_1(z)) - \rho(f_2(z + \varepsilon)) + \rho(f_2(z))\big) \\
&= \text{sign}\big(\rho(f_1(z + \varepsilon)) - \rho(f_2(z + \varepsilon))\big),
\end{aligned}$$
where the last equality uses the assumption $\rho(f_1(z)) \geq \rho(f_2(z))$ in *(ii)*. Therefore, we have
$$\lim_{\varepsilon \to 0^+} \text{sign}\big(\rho(f_1(z + \varepsilon)) - \rho(f_2(z + \varepsilon))\big) = +$$
and this completes the proof. $\square$

**Lemma 12.** *Let $\Psi$ be a network satisfying Condition 2, and consider $w \in \mathbb{R}^W$, $B \in \mathbb{N}$, $X \in \mathbb{R}^{N_0 \times B}$. For each $l \in [L]$, let $s_{l,j',b'} \in \{-, +\}$ for all $(j', b') \in [M_l] \times [B]$. For each $l \in [L]$ and $(i, b) \in [N_l] \times [B]$,*

- *let $\mathcal{I}_{l,i,b} \subset [M_l] \times [B]$ be the set of indices that $(\phi_l)_{i,b}$ uses (i.e., $z_{l,i,b}(w') = \max\{\rho_l(y_{l,j',b'}(w')) : (j', b') \in \mathcal{I}_{l,i,b}\}$ for all $w' \in \mathbb{R}^W$);*
- *let $(\mu_{l,i,b}, \nu_{l,i,b}) \in \mathcal{I}_{l,i,b}$ be the index that AD uses when computing $D^{\text{AD}}\sigma_{l,i,b}(y_l(w))$ (see Appendix B for details); and let $\mathcal{J}_{l,i,b} \subset \mathcal{I}_{l,i,b} \setminus \{(\mu_{l,i,b}, \nu_{l,i,b})\}$.*

*Here, $(\mu_{l,i,b}, \nu_{l,i,b})$ and $\mathcal{J}_{l,i,b}$ depend on $w$ which is fixed in this lemma. For each $l \in [L]$, let $\mathcal{N}_l = \{(j', b') \in [M_l] \times [B] : y_{l,j',b'}(w) \in \text{ndf}(\rho_l)\}$. Suppose that we are given $\zeta_1 \in \mathbb{R}^{W_1}, \ldots, \zeta_L \in \mathbb{R}^{W_L}$ such that for all $l \in [L]$ and $(i, b) \in [N_l] \times [B]$,*

$$\lim_{\varepsilon \to 0^+} \text{sign}\Big(\tau_{l,j',b'}(z_{0:l-1}(w), w_l + \varepsilon\zeta_l) - \tau_{l,j',b'}(z_{0:l-1}(w), w_l)\Big) = s_{l,j',b'}$$
$$\text{for all } (j', b') \in \mathcal{N}_l, \tag{7}$$

$$\lim_{\varepsilon \to 0^+} \text{sign}\Big(\rho_l\big(\tau_{l,\mu_{l,i,b},\nu_{l,i,b}}(z_{0:l-1}(w), w_l + \varepsilon\zeta_l)\big) - \rho_l\big(\tau_{l,j'',b''}(z_{0:l-1}(w), w_l + \varepsilon\zeta_l)\big)\Big) = +$$
$$\text{for all } (j'', b'') \in \mathcal{J}_{l,i,b}. \tag{8}$$

*Then, there exists a sequence $\{\eta_n\}_{n \in \mathbb{N}} \subset \mathbb{R}^W$ such that for each $l \in [L]$ and $(i, b) \in [N_l] \times [B]$,*

- $\lim_{n\to\infty} \eta_n = \mathbf{0}_W$,
- $y_{l,j',b'}(w + \eta_n) \in \mathrm{int}(\mathcal{P}(\rho_l, y_{l,j',b'}(w), s_{l,j',b'}))$ *for all* $(j', b') \in [M_l] \times [B]$ *and* $n \in \mathbb{N}$, *and*
- $\rho_l(y_{l,\mu_{l,i,b},\nu_{l,i,b}}(w + \eta_n)) > \rho_l(y_{l,j'',b''}(w + \eta_n))$ *for all* $(j'', b'') \in \mathcal{J}_{l,i,b}$ *and* $n \in \mathbb{N}$.

*Here,* $\mathcal{P}(f, x, +)$ *and* $\mathcal{P}(f, x, -)$ *were defined in Appendix A.*

*Proof of Lemma 12.* Without loss of generality, suppose that $\|\zeta_l\|_2 = 1$ for all $l \in [L]$. By Eqs. (7) and (8), we can choose $\delta_1, \ldots, \delta_L > 0$ such that for all $l \in [L]$, $(i, b) \in [N_l] \times [B]$, $(j', b') \in \mathcal{N}_l$, and $(j'', b'') \in \mathcal{J}_{l,i,b}$,

$$\mathrm{sign}\Big(\tau_{l,j',b'}(z_{0:l-1}(w), w_l + \varepsilon\zeta_l) - \tau_{l,j',b'}(z_{0:l-1}(w), w_l)\Big) = s_{l,j',b'}, \tag{9}$$

$$\rho_l\Big(\tau_{l,\mu_{l,i,b},\nu_{l,i,b}}(z_{0:l-1}(w), w_l + \varepsilon\zeta_l)\Big) > \rho_l\Big(\tau_{l,j'',b''}(z_{0:l-1}(w), w_l + \varepsilon\zeta_l)\Big) \tag{10}$$

for all $\varepsilon \in (0, \delta_l)$. Without loss of generality, we assume $\delta_1, \ldots, \delta_L$ also satisfy the following: for all $l \in [L]$ and $(j', b') \in [M_l] \times [B]$,

$$\tau_{l,j',b'}(z_{0:l-1}(w), w_l + \varepsilon\zeta_l) \in \mathrm{int}\Big(\mathcal{P}(\rho_l, y_{l,j',b'}(w), s_{l,j',b'})\Big) \tag{11}$$

for all $\varepsilon \in (0, \delta_l)$. We can always find such small enough $\delta_1, \ldots, \delta_L$ from Eq. (9), the continuity of $\tau_l$, and $\mu_1(\mathcal{P}(\rho_l, y_{l,j',b'}(w), s_{l,j',b'})) > 0$ (which holds by the piecewise-analyticity of $\rho_l$ and Definition 1).

We now claim that there exists $\{\eta_n\}_{n\in\mathbb{N}} \subset \mathbb{R}^W$ such that for any $n \in \mathbb{N}$, $l \in [L]$, and $(i, b) \in [N_l] \times [B]$,

(i) $\|\eta_n\|_2 \le \sqrt{1/n}$,
(ii) $y_{l,j',b'}(w + \eta_n) \in \mathrm{int}(\mathcal{P}(\rho_l, y_{l,j',b'}(w), s_{l,j',b'}))$ for all $(j', b') \in [M_l] \times [B]$,
(iii) $\rho_l(y_{l,\mu_{l,i,b},\nu_{l,i,b}}(w + \eta_n)) > \rho_l(y_{l,j'',b''}(w + \eta_n))$ for all $(j'', b'') \in \mathcal{J}_{l,i,b}$.

Then, $\{\eta_n\}_n$ is a desired sequence. To show the existence of such $\{\eta_n\}_n$, we construct $\{\eta_{n,k}\}_{n\in\mathbb{N}} \subset \mathbb{R}^W$ for each $k \in [L]$ such that for any $n \in \mathbb{N}$, $l \in [L] \setminus [L - k]$, and $(i, b) \in [N_l] \times [B]$,

$k$-(i) $(\eta_{n,k})_t = 0$ for all $t \in [W_1 + \cdots + W_{L-k}]$,
$k$-(ii) $\|\eta_{n,k}\|_2 \le \sqrt{k/(nL)}$,
$k$-(iii) $y_{l,j',b'}(w + \eta_{n,k}) \in \mathrm{int}(\mathcal{P}(\rho_l, y_{l,j',b'}(w), s_{l,j',b'}))$ for all $(j', b') \in [M_l] \times [B]$,
$k$-(iv) $\rho_l(y_{l,\mu_{l,i,b},\nu_{l,i,b}}(w + \eta_{n,k}))) > \rho_l(y_{l,j'',b''}(w + \eta_{n,k}))$ for all $(j'', b'') \in \mathcal{J}_{l,i,b}$.

Then, choosing $\eta_n = \eta_{n,L}$ completes the proof.

We construct such $\{\eta_{n,k}\}_n$ by induction on $k$. Consider the base case: $k = 1$. For each $n$, we choose $\eta_{n,1} = \mathbf{0}_{W_1 + \cdots + W_{L-1}} \oplus (\varepsilon_n \zeta_L)$ for $\varepsilon_n = \min\{\delta_L/2, \sqrt{1/(nL)}\}$. Then, $\eta_{n,1}$ satisfies 1-$\{(i), \ldots, (iv)\}$ by Eqs. (10) and (11) and by our choice of $\eta_{n,1}$ and $\varepsilon_n$. Now, consider a general $k > 1$. By the induction hypothesis on $k - 1$, there exists $\{\eta_{n,k-1}\}_n$ satisfying $(k - 1)$-$\{(i), \ldots, (iv)\}$. For each $n$, we choose $\eta_{n,k} = \eta_{n,k-1} + \mathbf{0}_{W_1 + \cdots + W_{L-k}} \oplus (\varepsilon'_n \zeta_{L-k+1}) \oplus \mathbf{0}_{W_{L-k+2} + \cdots + W_L}$ for some $\varepsilon'_n > 0$ so that $\eta_{n,k}$ satisfies $k$-$\{(i), \ldots, (iv)\}$. Such $\varepsilon'_n$ always exists as follows: for any $0 < \varepsilon'_n \le \min\{\delta_{L-k+1}/2, \sqrt{1/(nL))}\}$, $\eta_{n,k}$ satisfies

- $k$-$\{(i), (ii)\}$ and
- $k$-$\{(iii), (iv)\}$ for $l = L - k + 1$.

This is because $\eta_{n,k-1}$ satisfies $(k - 1)$-$\{(i), (ii)\}$ and Eqs. (10) and (11) hold. Furthermore, for a small enough $\varepsilon'_n > 0$, $\eta_{n,k}$ satisfies $k$-$\{(iii), (iv)\}$ for all $l > L - k + 1$, since $\eta_{n,k-1}$ satisfies $(k - 1)$-$\{(iii), (iv)\}$ and $\tau_{l'}, \sigma_{l'}$ are continuous for all $l'$. This completes the proof of Lemma 12. $\square$

**Lemma 13.** *Let $\Psi$ be a neural network satisfying Condition 2 with only trivial (or differentiable) maxpools, $w \in \mathbb{R}^W$, $B \in \mathbb{N}$, and $X \in \mathbb{R}^{N_0 \times B}$. For every $l \in [L]$, let $\{\mathcal{A}_{l,i}\}_{i \in [n_l]}$ be a partition of $\rho_l$ described in Definition 1. Suppose that for each $l \in [L]$, $j \in [M_l]$, and $b \in [B]$, there exists $k \in [n_l]$ such that $y_{l,j,b}(w; X) \in \mathrm{int}(\mathcal{A}_{l,k})$. Then, $\Psi(\,\cdot\,; X)$ is differentiable at $w$ (i.e., $D\Psi(w; X)$ exists) and $D\Psi(w; X) = D^{\mathtt{AD}}\Psi(w; X)$.*

*Proof of Lemma 13.* By Definition 1, the condition $y_{l,j,b}(w; X) \in \mathrm{int}(\mathcal{A}_{l,k})$ implies that $\rho_l$ is differentiable at $y_{l,j,b}(w; X)$ for all $l, j, b$. Since $\sigma_l$ uses a trivial (or differentiable) maxpool, $\sigma_l$ is differentiable at $y_l(w; X)$ for all $l$. Moreover, $\tau_l$ is differentiable on its domain for all $l$ by Condition 2. Hence, $\Psi(\,\cdot\,; X)$ is differentiable at $w$ by the definition of $\Psi$ and the chain rule for differentiation. Further, we have $D\Psi(w; X) = D^{\mathtt{AD}}\Psi(w; X)$ by the definition of $D^{\mathtt{AD}}\Psi$ (Appendix B) and by the chain rule with $D\rho_l(y_{l,j,b}(w)) = D^{\mathtt{AD}}\rho_l(y_{l,j,b}(w))$ for all $l, j, b$. $\qquad\square$

**Lemma 14.** *Let $\Psi$ be a neural network satisfying Condition 2. Consider any $l \in [L]$, $w \in \mathbb{R}^W$, and $X \in \mathbb{R}^{N_0 \times B}$. Then, for any $f_1, \ldots, f_{l-1}, f_{l+1}, \ldots, f_L : \mathbb{R} \to \mathbb{R}$, and for any $K \in \mathbb{N}$, $f_{l,1}, \ldots, f_{l,K} : \mathbb{R} \to \mathbb{R}$, and $c_1, \ldots, c_K \in \mathbb{R}$ with $c_1 + \cdots + c_K = 1$, we have*

$$D^{\mathtt{AD}}_{(f_1, \ldots, f_{l-1}, \sum_{i \in [K]} c_i f_{l,i}, f_{l+1}, \ldots, f_L)} \Psi(w; X) = \sum_{i \in [K]} c_i D^{\mathtt{AD}}_{(f_1, \ldots, f_{l-1}, f_{l,i}, f_{l+1}, \ldots, f_L)} \Psi(w; X).$$

*Proof of Lemma 14.* This follows from the definition of AD (Appendix B). We omit the proof as it uses a simple induction over the definition of $D^{\mathtt{AD}}\Psi(w; X)$. We remark that the condition $c_1 + \cdots + c_K = 1$ is essential: this lemma no longer holds without the condition. $\qquad\square$

**Lemma 15.** *Let $\Psi$ be a neural network satisfying Condition 2, $K_1, \ldots, K_L \in \mathbb{N}$, and $f_{l,k} : \mathbb{R} \to \mathbb{R}$ for all $l \in [L]$ and $k \in [K_l]$. Let $\mathcal{G} = \{(g_1, \ldots, g_L) : g_l \in \{f_{l,1}, \ldots, f_{l,K_l}\} \text{ for all } l \in [L]\}$. Then, for any $h = (h_1, \ldots, h_L)$ such that $h_l \in \mathrm{conv}(\{f_{l,1}, \ldots, f_{l,K_l}\})$ for all $l \in [L]$, it holds that*

$$D^{\mathtt{AD}}_h \Psi(w; X) \in \mathrm{conv}(\{D^{\mathtt{AD}}_g \Psi(w; X) : g \in \mathcal{G}\}).$$

*Proof of Lemma 15.* We show the following claim using induction on $n$: for all $n \in [L]$, if $h = (h_1, \ldots, h_L)$ satisfies that $h_l \in \mathrm{conv}(\{f_{l,1}, \ldots, f_{l,K_l}\})$ for all $l \leq n$ and $h_l \in \{f_{l,1}, \ldots, f_{l,K_l}\}$ for all $l > n$, then
$$D^{\mathtt{AD}}_h \Psi(w; X) \in \mathrm{conv}(\{D^{\mathtt{AD}}_g \Psi(w; X) : g \in \mathcal{G}\}).$$
Then, the desired statement follows from the case for $n = L$.

The base case (i.e., when $n = 1$) follows directly from Lemma 14 with $l = 1$. Consider a general case (i.e., when $n \geq 2$) and let $h = (h_1, \ldots, h_L)$ such that $h_l \in \mathrm{conv}(\{f_{l,1}, \ldots, f_{l,K_l}\})$ for all $l \leq n$ and $h_l \in \{f_{l,1}, \ldots, f_{l,K_l}\}$ for all $l > n$. Let $\ell_k = (h_1, \ldots, h_{n-1}, f_{n,k}, h_{n+1}, \ldots, h_L)$ for all $k \in [K_n]$. Then, by the induction hypothesis on $n - 1$, we have

$$D^{\mathtt{AD}}_{\ell_k} \Psi(w; X) \in \mathrm{conv}(\{D^{\mathtt{AD}}_g \Psi(w; X) : g \in \mathcal{G}\})$$

for all $k \in [K_n]$. Since $h_n \in \mathrm{conv}(\{f_{n,1}, \ldots, f_{n,K_n}\})$, the definition of $h$ and Lemma 14 imply

$$D^{\mathtt{AD}}_h \Psi(w; X) \in \mathrm{conv}(\{D^{\mathtt{AD}}_{\ell_k} \Psi(w; X) : k \in [K_n]\}) \subset \mathrm{conv}(\{D^{\mathtt{AD}}_g \Psi(w; X) : g \in \mathcal{G}\}).$$

This completes the proof of Lemma 15. $\qquad\square$

# D  PROOFS OF RESULTS IN SECTION 3.1

## D.1  PROOF OF THEOREM 1

Fix $w \in \mathbb{R}^W$, $B = 1$, and $X \in \mathbb{R}^{N_0 \times B}$.

**Case $\gamma = \text{B}$.**  First, choose $s_{l,i} \in \{-, +\}$ for each $l \in [L]$ and $i \in [N_l]$ so that

$$D^{\text{AD}} \rho_l(y_{l,i}(w)) = D^{s_{l,i}} \rho_l(y_{l,i}(w)). \tag{12}$$

Such $s_{l,i}$ always exists as we assumed $D^{\text{AD}} \rho_l(x) \in \partial^{\text{B}} \rho_l(x)$ for all $x \in \text{ndf}(\rho_l)$ and $\rho_l$ is piecewise-analytic. Let $s_l = (s_{l,1}, \ldots, s_{l,N_l})$, and let $\zeta_l = \mathbf{0}_{W_l - N_l} \oplus (s_{l,1}, \ldots, s_{l,N_l}) \in \mathbb{R}^{W_l}$ for all $l \in [L]$, i.e., the coordinates corresponding to the bias parameters in $\zeta_l$ have non-zero values $s_{l,1}, \ldots, s_{l,N_l}$. Then, one can observe that

$$\lim_{\varepsilon \to 0^+} \text{sign}\Big( \tau_l(z_{0:l-1}(w), w_l + \varepsilon \zeta_l) - \tau_l(z_{0:l-1}(w), w_l) \Big) = s_l$$

for all $l \in [L]$. Then, by Lemma 12, there exists $\{\eta_n\}_n \subset \mathbb{R}^W$ such that for all $l \in [L]$ and $i \in [N_l]$,

*(i)* $\eta_n \to \mathbf{0}_W$ as $n \to \infty$ and

*(ii)* $y_{l,i}(w + \eta_n) \in \text{int}(\mathcal{P}(\rho_l, y_{l,i}(w), s_{l,i}))$ for all $n \in \mathbb{N}$.

From these, it holds that

$$D\Psi(w + \eta_n) = D^{\text{AD}}\Psi(w + \eta_n) \to D^{\text{AD}}\Psi(w) \qquad \text{as } n \to \infty. \tag{13}$$

Here, the existence of the first term (i.e., $\Psi$ is differentiable at $w + \eta_n$) and the first equality are both by Lemma 13 and *(ii)*, and the convergence is by the definition of $D^{\text{AD}}\Psi$ (Appendix B), Eq. (12), and *(i)*. By combining Eq. (13) and *(i)*, we have $D^{\text{AD}}\Psi(w) \in \partial^{\text{B}}\Psi(w)$ as desired.

**Case $\gamma = \text{C}$.**  Let $\text{ndf}(\rho_l) = \{x_{l,1}, \ldots, x_{l,k_l}\}$ for all $l \in [L]$, and let $\mathcal{T} = \prod_{l \in [L]} \{-, +\}^{k_l}$. For each $l \in [L]$ and $t \in \mathcal{T}$ with $t = (t_1, \ldots, t_L)$ and $t_l \in \{-, +\}^{k_l}$, we define $h_l, f_{l,t} : \mathbb{R} \to \mathbb{R}$ as

$$h_l(x) = D^{\text{AD}} \rho_l(x), \qquad f_{l,t}(x) = \begin{cases} D\rho_l(x) & \text{if } x \notin \text{ndf}(\rho_l) \\ D^{t_{l,i}} \rho_l(x) & \text{if } x = x_{l,i}. \end{cases}$$

Then, the assumption $D^{\text{AD}} \rho_l(x) \in \partial^{\text{C}} \rho_l(x)$ implies that for all $l \in [L]$ and $x \in \mathbb{R}$,

$$h_l(x) = \begin{cases} D\rho_l(x) & \text{if } x \notin \text{ndf}(\rho_l) \\ \lambda_{l,i} D^- \rho_l(x) + (1 - \lambda_{l,i}) D^+ \rho_l(x) & \text{if } x = x_{l,i} \end{cases}$$

for some $\lambda_{l,i} \in [0, 1]$. From this, one can observe that

$$h_l \in \text{conv}(\{f_{l,t} : t \in \mathcal{T}\}) \quad \text{for all } l \in [L].$$

Using this observation, we can derive the desired conclusion:

$$\begin{aligned} D^{\text{AD}}\Psi(w) &= D^{\text{AD}}_{(h_1, \ldots, h_L)}\Psi(w) \\ &\in \text{conv}\big(\big\{ D^{\text{AD}}_{(g_1, \ldots, g_L)}\Psi(w) : g_l \in \{f_{l,t} : t \in \mathcal{T}\} \text{ for all } l \in [L] \big\}\big) \\ &\subseteq \text{conv}\big(\partial^{\text{B}}\Psi(w)\big) = \partial^{\text{C}}\Psi(w), \end{aligned}$$

where the first inclusion is by Lemma 15, and the second inclusion follows from the fact that $D^{\text{AD}}_{(g_1, \ldots, g_L)}\Psi(w) \in \partial^{\text{B}}\Psi(w)$ for any $g_l \in \{f_{l,t} : t \in \mathcal{T}\}$ (which holds by $f_{l,t}(x) \in \partial^{\text{B}}\rho_l(x)$ and our proof for the previous case $\gamma = \text{B}$). This completes the proof of Theorem 1.

## D.2  PROOF OF LEMMA 2

Consider a neural network $\Psi$ defined as follows:

- $B = 1$, $L = 1$, $N_0 = 1$, $N_1 = 2$, and $W_1 = 1$.
- $\tau_1 : \mathbb{R}^{N_0 \times B} \times \mathbb{R}^{W_1} \to \mathbb{R}^{N_1 \times B}$ is defined as $\tau_1(X, w_1) = (Xw_1, -Xw_1)$.

- $\rho_1 : \mathbb{R} \to \mathbb{R}$ is defined as $\rho_1 = \mathrm{ReLU}$ with $D^{\mathtt{AD}}\rho_1(0) = 0$.
- $\ell : \mathbb{R}^{N_1 \times B} \to \mathbb{R}$ is defined as $\ell(x_1, x_2) = x_1 - x_2$.

Then, $\Psi$ is expressed as $\Psi(w; X) = \ell(\sigma_1(\tau_1(X, w_1)))$, where $w = w_1 \in \mathbb{R}$, $X \in \mathbb{R}$, and $\sigma_1 = (\rho_1, \rho_1)$. Moreover, $\Psi$ satisfies Condition 1 without distinct bias parameters, and $D^{\mathtt{AD}}\rho_1(x) \in \partial^{\mathtt{B}}\rho_1(x)$ for all $x \in \mathsf{ndf}(\rho_1) = \{0\}$. We now consider $X^* = 1$. Then, for all $w = w_1 \in \mathbb{R}$,

$$\Psi(w; X^*) = \mathrm{ReLU}(w_1) - \mathrm{ReLU}(-w_1) = w_1,$$

so $\partial^{\mathtt{C}}\Psi(0; X^*) = \{1\}$. However, $D^{\mathtt{AD}}\Psi(0; X^*) = 0 \notin \partial^{\mathtt{C}}\Psi(0; X^*)$. This completes the proof of Lemma 2.

### D.3 PROOF OF LEMMA 3

Consider a neural network $\Psi$ defined as follows:

- $B = 2$, $L = 1$, $N_0 = N_1 = 1$, and $W_1 = 1$.
- $\tau_1 : \mathbb{R}^{N_0 \times B} \times \mathbb{R}^{W_1} \to \mathbb{R}^{N_1 \times B}$ is defined as $\tau_1(X, w_1) = X + w_1$.
- $\rho_1 : \mathbb{R} \to \mathbb{R}$ is defined as $\rho_1 = \mathrm{HardSigmoid}$ with $D^{\mathtt{AD}}\rho_1(-3) = D^{\mathtt{AD}}\rho_1(3) = 0$ (see Footnote 3 for the definition of HardSigmoid).
- $\ell : \mathbb{R}^{N_1 \times B} \to \mathbb{R}$ is defined as $\ell(x_1, x_2) = x_1 + x_2$.

Then, $\Psi$ is expressed as $\Psi(w; X) = \ell(\sigma_1(\tau_1(X, w_1)))$, where $w = w_1 \in \mathbb{R}$, $X \in \mathbb{R}^2$, and $\sigma_1 = (\rho_1, \rho_1)$. Moreover, $\Psi$ satisfies Condition 1 with distinct bias parameters, and $D^{\mathtt{AD}}\rho_1(x) \in \partial^{\mathtt{B}}\rho_1(x)$ for all $x \in \mathsf{ndf}(\rho_1) = \{-3, 3\}$. We now consider $X^* = (3, -3)$. Then, for all $w = w_1 \in \mathbb{R}$,

$$\Psi(w; X^*) = \mathrm{HardSigmoid}(3 + w_1) + \mathrm{HardSigmoid}(-3 + w_1) = 1 + w_1/6,$$

so $\partial^{\mathtt{C}}\Psi(0; X^*) = \{1/6\}$. However, $D^{\mathtt{AD}}\Psi(0; X^*) = 0 \notin \partial^{\mathtt{C}}\Psi(0; X^*)$. This completes the proof of Lemma 3.

### D.4 PROOF OF THEOREM 4

Theorem 4 is a special case of Theorem 6, and the proofs of the two theorems are almost identical. Hence, we omit the proof of Theorem 4; the proof of Theorem 6 can be found in Section E.1.

### D.5 PROOF OF THEOREM 5

Fix $w \in \mathbb{R}^W$, $B \in \mathbb{N}$, and $X \in \mathbb{R}^{N_0 \times B}$. For each $l \in [L]$, let $\mathcal{N}_l = \{(i, b) \in [N_l] \times [B] : y_{l,i,b}(w) \in \mathsf{ndf}(\rho_l)\}$. We note that by the assumption, $\mathcal{N}_l \neq \emptyset$ implies $\mathrm{rank}(z_{l-1}(w)) = B$ for any $l$.

**Case $\gamma = \mathtt{B}$.** First, we choose $s_{l,i,b} \in \{-, +\}$ for each $l \in [L]$ and $(i, b) \in [N_l] \times [B]$ such that

$$D^{\mathtt{AD}}\rho_l(y_{l,i,b}(w)) = D^{s_{l,i,b}}\rho_l(y_{l,i,b}(w)) \qquad \text{if } (i, b) \in \mathcal{N}_l; \tag{14}$$

we can choose an arbitrary $s_{l,i,b} \in \{-, +\}$ (e.g., $s_{l,i,b} = +$) if $(i, b) \notin \mathcal{N}_l$. Such $s_{l,i,b}$ always exists since $D^{\mathtt{AD}}\rho_l(x) \in \partial^{\gamma}\rho_l(x) = \partial^{\mathtt{B}}\rho_l(x)$ for all $x \in \mathsf{ndf}(\rho_l)$ (by the assumption) and since $\rho_l$ is piecewise-analytic. We then choose $\varphi_{l,i} \in \mathbb{R}^{1 \times N_{l-1}}$ for each $l \in [L]$ and $i \in [N_l]$ such that

$$\varphi_{l,i} z_{l-1}(w) = [s_{l,i,b}]_b \qquad \text{if } \mathcal{N}_l \neq \emptyset; \tag{15}$$

we can choose an arbitrary $\varphi_{l,i}$ if $\mathcal{N}_l = \emptyset$. Such $\varphi_{l,i}$ always exists since $\mathcal{N}_l \neq \emptyset$ implies $\mathrm{rank}(z_{l-1}(w)) = B$ as noted above.

Next, let $\Phi_l \in \mathbb{R}^{N_l \times N_{l-1}}$ be a matrix whose $i$-th row is $\varphi_{l,i}$. Also, let $\zeta_l = \mathrm{vec}(\Phi_l)$ if the $l$-th layer does not have bias parameters, and $\zeta_l = \mathrm{vec}(\Phi_l) \oplus \mathbf{0}_{N_l}$ otherwise. Then, by Eq. (15),

$$\Phi_l z_{l-1}(w) = [s_{l,i,b}]_{i,b} \qquad \text{for all } l \in [L] \text{ with } \mathcal{N}_l \neq \emptyset. \tag{16}$$

Under this setup, we claim that for all $l \in [L]$ and $(i, b) \in \mathcal{N}_l$,

$$\lim_{\varepsilon \to 0^+} \mathrm{sign}\Big(\tau_{l,i,b}(z_{0:l-1}(w), w_l + \varepsilon\zeta_l) - \tau_{l,i,b}(z_{0:l-1}(w), w_l)\Big) = s_{l,i,b}. \tag{17}$$

This claim holds as follows: for any $l \in [L]$ with $\mathcal{N}_l \neq \emptyset$, if the $l$-th layer has bias parameters, then

$$
\begin{aligned}
& \tau_{l,i,b}\Big(z_{0:l-1}(w), w_l + \varepsilon\zeta_l\Big) - \tau_{l,i,b}\Big(z_{0:l-1}(w), w_l\Big) \\
&= \tau_{l,i,b}\Big(z_{0:l-1}(w), (\mathsf{vec}(A_l) \oplus b_l) + \varepsilon(\mathsf{vec}(\Phi_l) \oplus \mathbf{0}_{N_l})\Big) - \tau_{l,i,b}\Big(z_{0:l-1}(w), \mathsf{vec}(A_l) \oplus b_l\Big) \\
&= \tau_{l,i,b}\Big(z_{0:l-1}(w), \mathsf{vec}(A_l + \varepsilon\Phi_l) \oplus b_l\Big) - \tau_{l,i,b}\Big(z_{0:l-1}(w), \mathsf{vec}(A_l) \oplus b_l\Big) \\
&= \Big((A_l + \varepsilon\Phi_l)z_{l-1}(w) + b_l\Big) - \Big(A_l z_{l-1}(w) + b_l\Big) \\
&= \varepsilon \cdot \Phi_l z_{l-1}(w) = \varepsilon \cdot [s_{l,i,b}]_{i,b},
\end{aligned}
$$

where the first and third equalities are by the assumption, and the last equality is by Eq. (16). We can use a similar argument to prove the case when the $l$-th layer does not have bias parameters.

Finally, by Lemma 12 applied to Eq. (17), there exists $\{\eta_n\}_n \subset \mathbb{R}^W$ such that for each $l \in [L]$, $i \in [N_l]$, and $b \in [B]$,

*(i)* $\eta_n \to \mathbf{0}_W$ as $n \to \infty$ and

*(ii)* $y_{l,i,b}(w + \eta_n) \in \mathsf{int}(\mathcal{P}(\rho_l, y_{l,i,b}(w), s_{l,i,b}))$ for all $n \in \mathbb{N}$.

From these, it holds that

$$D\Psi(w + \eta_n) = D^{\mathtt{AD}}\Psi(w + \eta_n) \to D^{\mathtt{AD}}\Psi(w) \qquad \text{as } n \to \infty. \tag{18}$$

Here, the existence of the first term (i.e., $\Psi$ is differentiable at $w + \eta_n$) and the first equality are both by Lemma 13 and *(ii)*, and the convergence is by the definition of $D^{\mathtt{AD}}\Psi$ (Appendix B), Eq. (14), and *(i)*. By combining Eq. (18) and *(i)*, we have $D^{\mathtt{AD}}\Psi(w) \in \partial^{\mathtt{B}}\Psi(w)$ as desired.

**Case $\gamma = \mathtt{C}$.** The proof for this case is identical to the proof of Theorem 1 for the same case, except that we now rely on the proof for the case $\gamma = \mathtt{B}$ in this theorem (not in Theorem 1). This completes the proof of Theorem 5.

# E   PROOFS OF RESULTS IN SECTION 3.2

## E.1   PROOF OF THEOREM 6

Since all maxpools are trivial, we have $N_l = M_l$ for all $l$.

**Case $\lambda_1, \ldots, \lambda_L \in \{0, 1\}$.** Let $s_l = -$ if $\lambda_l = 1$ and $s_l = +$ if $\lambda_l = 0$ for all $l \in [L]$. Let $\zeta_l = \mathbf{0}_{W_l - C_l} \oplus (s_l \mathbf{1}_{C_l})$ for all $l \in [L]$. Then, one can observe that

$$\lim_{\varepsilon \to 0^+} \operatorname{sign}\Big(\tau_l(z_{0:l-1}(w), w_l + \varepsilon \zeta_l) - \tau_l(z_{0:l-1}(w), w_l)\Big) = s_l \mathbf{1}_{N_l \times B}$$

for all $l \in [L]$. Then, by Lemma 12, there exists $\{\eta_n\}_n \subset \mathbb{R}^W$ such that for each $l \in [L]$, $i \in [N_l]$, and $b \in [B]$,

*(i)* $\eta_n \to \mathbf{0}_W$ as $n \to \infty$ and
*(ii)* $y_{l,i,b}(w + \eta_n) \in \operatorname{int}(\mathcal{P}(\rho_l, y_{l,i,b}(w), s_l))$ for all $n \in \mathbb{N}$.

By Lemma 13, *(i)*–*(ii)*, the definition of $D^{\mathsf{AD}}\Psi(w)$, and by $D^{\mathsf{AD}}\rho_l = D^{s_l}\rho_l$, we have $D\Psi(w + \eta_n) = D^{\mathsf{AD}}\Psi(w + \eta_n) \to D^{\mathsf{AD}}\Psi(w)$ as $n \to \infty$ (for a more detailed argument, refer to Eq. (13) and the text below that). From this and *(i)*, it holds that $D^{\mathsf{AD}}\Psi(w) \in \partial^{\mathsf{B}}\Psi(w)$.

**Case $\lambda_1, \ldots, \lambda_L \in [0, 1]$.** For $t = (t_1, \ldots, t_L) \in \{-, +\}^L$, let $f_t = (f_{1,t}, \ldots, f_{L,t})$ where

$$f_{l,t}(x) = \begin{cases} D\rho_l(x) & \text{if } x \notin \operatorname{ndf}(\rho_l) \\ D^{t_l}\rho_l(x) & \text{if } x \in \operatorname{ndf}(\rho_l) \end{cases}$$

for all $l \in [L]$. Also, let $h = (h_1, \ldots, h_L)$ with $h_l = D^{\mathsf{AD}}\rho_l$. By the assumption, we know that

$$D^{\mathsf{AD}}\rho_l(x) = \begin{cases} D\rho_l(x) & \text{if } x \notin \operatorname{ndf}(\rho_l) \\ \lambda_l D^-\rho_l(x) + (1 - \lambda_l)D^+\rho_l(x) & \text{if } x \in \operatorname{ndf}(\rho_l) \end{cases}$$

for some $\lambda_l \in [0, 1]$. From this, we can observe that $h_l \in \operatorname{conv}(\{f_{l,t} : t \in \{-, +\}^L\})$ for all $l \in [L]$. From this, we obtain the desired conclusion:

$$D^{\mathsf{AD}}\Psi(w) = D_h^{\mathsf{AD}}\Psi(w) \in \operatorname{conv}\Big(\big\{D_{(g_1, \ldots, g_L)}^{\mathsf{AD}}\Psi(w) : g_l \in \{f_{l,t} : t \in \{-, +\}^L\} \text{ for all } l \in [L]\big\}\Big)$$
$$\subseteq \operatorname{conv}\big(\partial^{\mathsf{B}}\Psi(w)\big) = \partial^{\mathsf{C}}\Psi(w),$$

where the first inclusion is by Lemma 15 and the second inclusion is by the observation that $D_{(g_1, \ldots, g_L)}^{\mathsf{AD}}\Psi(w) \in \partial^{\mathsf{B}}\Psi(w)$ for any $g_l \in \{f_{l,t} : t \in \{-, +\}^L\}$ (which follows from $f_{l,t} \in \{D^-\rho_l, D^+\rho_l\}$ and our proof for the previous case). This completes the proof of Theorem 6.

## E.2   PROOF OF LEMMA 7

Consider a neural network $\Psi$ defined as follows:

- $B = 1$, $L = 1$, $N_0 = 1$, $M_1 = 4$, $N_1 = 2$, and $W_1 = 2$.
- $\tau_1 : \mathbb{R}^{N_0 \times B} \times \mathbb{R}^{W_1} \to \mathbb{R}^{M_1 \times B}$ is defined as $\tau_1(X, (u_1, b_1)) = (Xu_1 + b_1, b_1, -Xu_1 + b_1, b_1)$.
- $\rho_1 : \mathbb{R} \to \mathbb{R}$ is the identity function.
- $\phi_1 : \mathbb{R}^{M_1 \times B} \to \mathbb{R}^{N_1 \times B}$ is defined as $\phi_1(x_1, x_2, x_3, x_4) = (\max_2(x_1, x_2), \max_2(x_3, x_4))$, where AD uses $D^{\mathsf{AD}}\max_2(x_1, x_2) = (1, 0)$ for all $x_1 = x_2$ (i.e., AD uses the first index of $\max_2$ if the two inputs are identical).
- $\ell : \mathbb{R}^{N_1 \times B} \to \mathbb{R}$ is defined as $\ell(x_1, x_2) = x_1 - x_2$.

Then, $\Psi$ is expressed as $\Psi(w; X) = \ell(\phi_1(\tau_1(X, (u_1, b_1))))$, where $w = (u_1, b_1) \in \mathbb{R}^2$ and $X \in \mathbb{R}$. Moreover, $\Psi$ satisfies Condition 2 with shared bias parameters and non-trivial maxpools, and $D^{\mathsf{AD}}\rho_1(x) \in D^-\rho_1(x)$ for all $x \in \operatorname{ndf}(\rho_1) = \emptyset$. We now consider $X^* = 1$. Then, for all $w = (u_1, b_1) \in \mathbb{R}^2$,

$$\Psi(w; X^*) = \max\{u_1 + b_1, b_1\} - \max\{-u_1 + b_1, b_1\} = u_1,$$

so $\partial^{\mathsf{C}}\Psi((0, 0); X^*) = \{(1, 0)\}$. However, $D^{\mathsf{AD}}\Psi((0, 0); X^*) = (2, 0) \notin \partial^{\mathsf{C}}\Psi((0, 0); X^*)$. This completes the proof of Lemma 7.

### E.3 PROOF OF THEOREM 8

In this proof, we write $\pi_i$, $\mathcal{T}_j$, and $\mathcal{S}_j$ appearing in Algorithm 1 as $\pi_{l,i}$, $\mathcal{T}_{l,j}$, and $\mathcal{S}_{l,j}$ to make their dependency on $l \in [L]$ explicit. Consider any $w \in \mathbb{R}^W$, $B \in \mathbb{N}$, and $X \in \mathbb{R}^{N_0 \times B}$. Suppose that $\mathcal{P}_l \neq \emptyset$ for all $l \in [L]$. Let

$$\mathcal{N}_l = \{(j', b') \in [M_l] \times [B] : y_{l,j',b'}(w) \in \mathsf{ndf}(\rho_l)\}$$

for each $l \in [L]$ as in the statement of Lemma 12.

**Step 1.** We choose $\zeta_l \in \mathcal{P}_l$ for each $l \in [L]$, and choose $s_{l,j',b'} \in \{-,+\}$ for each $l \in [L]$ and $(j',b') \in [M_l] \times [B]$ such that

$$D^{\mathtt{AD}}\rho_l(y_{l,j',b'}(w)) = D^{s_{l,j',b'}}\rho_l(y_{l,j',b'}(w)). \tag{19}$$

First, we claim that for each $l \in [L]$ and $(j',b') \in \mathcal{N}_l$,

$$\lim_{\varepsilon \to 0^+} \mathrm{sign}\Big(\tau_{l,j',b'}(z_{0:l-1}(w), w_l + \varepsilon\zeta_l) - \tau_{l,j',b'}(z_{0:l-1}(w), w_l)\Big) = s_{l,j',b'}. \tag{20}$$

To prove this, fix $l \in [L]$ and $(j',b') \in \mathcal{N}_l$, and define a function $f_{l,j',b'} : \mathbb{R} \to \mathbb{R}$ as $f_{l,j',b'}(\varepsilon) = \tau_{l,j',b'}(z_{0:l-1}(w), w_l + \varepsilon\zeta_l)$. Then,

$$\mathrm{sign}(Df_{l,j',b'}(0)) = \mathrm{sign}\left(\left\langle \frac{\partial \tau_{l,j',b'}}{\partial w_l}(z_{0:l-1}(w), w_l), \zeta_l \right\rangle\right) = \mathrm{sign}(\langle \pi_{l,j',b'}, \zeta_l \rangle) = s_{l,j',b'}, \tag{21}$$

where $\pi_{l,j',b'}$ corresponds to $\pi_{j',b'}$ in Algorithm 1 (as noted above), the second equality is by Line 4 in Algorithm 1, and the last equality is by Lines 6–7 in Algorithm 1, $\zeta_l \in \mathcal{P}_l$, and Eq. (19). This implies that $f_{l,j',b'}$ is a non-constant analytic function (since $Df_{l,j',b'}(0) \neq 0$). Hence, we can apply Lemma 10 to $f_{l,j',b'}$ and this yields Eq. (20).

Next, we claim that for all $l \in [L]$, $(i,b) \in [N_l] \times [B]$, and $(j',b') \in \mathcal{I}_{l,i,b} \setminus \mathcal{T}_{l,i,b}$,

$$\lim_{\varepsilon \to 0^+} \mathrm{sign}\Big(\rho_l\big(\tau_{l,\mu_{l,i,b},\nu_{l,i,b}}(z_{0:l-1}(w), w_l + \varepsilon\zeta_l)\big) - \rho_l\big(\tau_{l,j',b'}(z_{0:l-1}(w), w_l + \varepsilon\zeta_l)\big)\Big) = + \tag{22}$$

where $\mu_{l,i,b}, \nu_{l,i,b}, \mathcal{I}_{l,i,b}$ are defined as in Lemma 12 and $\mathcal{T}_{l,i,b}$ corresponds to $\mathcal{T}_{i,b}$ in Algorithm 1 (as noted above). To show this, fix $l \in [L]$ and $(i,b) \in [N_l] \times [B]$. Then, Eq. (22) clearly holds for all $(j',b') \in \mathcal{I}_{l,i,b} \setminus \mathcal{S}_{l,i,b}$ by Line 11 in Algorithm 1 and the continuity of $\rho_l$ and $\tau_l$, where $\mathcal{S}_{l,i,b}$ corresponds to $\mathcal{S}_{i,b}$ in the algorithm (as noted above). Also, we immediately obtain Eq. (22) for the remaining $(j',b')$ (i.e., $(j',b') \in \mathcal{S}_{l,i,b} \setminus \mathcal{T}_{l,i,b}$) by applying Lemma 11 to $\rho_l$, $f_{l,\mu_{l,i,b},\nu_{l,i,b}}$, and $f_{l,j',b'}$. Here, Lemma 11 is applicable because: the condition *(i)* of the lemma follows from Eqs. (19) and (21), and the condition *(ii)* of the lemma follows from the definition of $\mu_{l,i,b}, \nu_{l,i,b}$, Eq. (21), and Line 15 in Algorithm 1.

Finally, by applying Lemma 12 to Eqs. (20) and (22), we obtain the following: there exists a sequence $\{\eta_n\}_{n\in\mathbb{N}} \subset \mathbb{R}^W$ such that for each $l \in [L]$ and $(i,b) \in [N_l] \times [B]$,

*(i)* $\lim_{n\to\infty} \eta_n = \mathbf{0}_W$,

*(ii)* $y_{l,j',b'}(w + \eta_n) \in \mathrm{int}(\mathcal{P}(\rho_l, y_{l,j',b'}(w), s_{l,j',b'}))$ for all $(j',b') \in [M_l] \times [B]$ and $n \in \mathbb{N}$, and

*(iii)* $\rho_l(y_{l,\mu_{l,i,b},\nu_{l,i,b}}(w + \eta_n)) > \rho_l(y_{l,j',b'}(w + \eta_n))$ for all $(j',b') \in \mathcal{I}_{l,i,b} \setminus \mathcal{T}_{l,i,b}$ and $n \in \mathbb{N}$.

Let $\mathcal{E}'$ be a collection of all sequences $\{\eta'_n\}_n \subseteq \mathbb{R}^W$ that satisfy *(i)*–*(iii)*. Then, $\mathcal{E}'$ is not empty since $\{\eta_n\}_n \in \mathcal{E}'$.

**Step 2.** We first consider an easy case. That is, suppose that there is a sequence $\{\eta'_n\}_n \in \mathcal{E}'$ satisfying the following: for each $l, i, b$, there exists an index $(\mu^*_{l,i,b}, \nu^*_{l,i,b}) \in \mathcal{T}_{l,i,b}$ such that

$$\arg\max_{(j',b') \in \mathcal{I}_{l,i,b}} \rho_l(y_{l,j',b'}(w + \eta'_n)) = \{(\mu^*_{l,i,b}, \nu^*_{l,i,b})\} \qquad \text{for all } n \in \mathbb{N}. \tag{23}$$

We remark that $(\mu^*_{l,i,b}, \nu^*_{l,i,b})$ can be different from $(\mu_{l,i,b}, \nu_{l,i,b})$. Then, it holds that

$$D\Psi(w + \eta'_n) = D^{\mathtt{AD}}\Psi(w + \eta'_n) \to D^{\mathtt{AD}}\Psi(w) \qquad \text{as } n \to \infty. \tag{24}$$

The equality is by *(ii)* and Lemma 13. Here, the lemma is applicable because for each $n$, $\Psi$ is identical to a network with only differentiable maxpools, on some open neighborhood of $w + \eta'_n$; this follows from the existence of $(\mu^*_{l,i,b}, \nu^*_{l,i,b})$. The convergence in the above equation follows from the definition of $D^{\text{AD}}\Psi$ (see Appendix B), *(i)*–*(ii)*, Eq. (23), and two observations: for all $l, i, b$,

$$D^{s_{l,j',b'}}\rho_l(y_{l,j',b'}(w)) = D^{\text{AD}}\rho_l(y_{l,j',b'}(w)) \qquad \text{for all } (j', b') \in [M_l] \times [B],$$

$$\frac{\partial^{\text{AD}}\rho_l(y_{l,j',b'}(w))}{\partial w_{1:l}} = \frac{\partial^{\text{AD}}\rho_l(y_{l,j'',b''}(w))}{\partial w_{1:l}} \qquad \begin{array}{l} \text{for } (j', b') = (\mu^*_{l,i,b}, \nu^*_{l,i,b}) \text{ and} \\ (j'', b'') = (\mu_{l,i,b}, \nu_{l,i,b}), \end{array}$$

where the first equality is by the definition of $s_{l,j',b'}$ (see Eq. (19)) and the second equality is by $(\mu^*_{l,i,b}, \nu^*_{l,i,b}), (\mu_{l,i,b}, \nu_{l,i,b}) \in \mathcal{T}_{l,i,b}$ and the definition of $\mathcal{T}_{l,i,b}$ (see Line 14 of Algorithm 1). From Eq. (24) and *(i)*, we obtain the desired conclusion: $D^{\text{AD}}\Psi(w) \in \partial^{\text{B}}\Psi(w)$.

**Step 3.** We now consider a general case. That is, we no longer make the assumption considered in Step 2. For any $l \in [L]$ and $(i, b) \in [N_l] \times [B]$, we define the function $\mathcal{G}_{l,i,b} : \mathbb{R}^W \to 2^{\mathcal{I}_{l,i,b}}$ as

$$\mathcal{G}_{l,i,b}(\delta) \triangleq \underset{(j',b') \in \mathcal{I}_{l,i,b}}{\arg\max} \ \rho_l(y_{l,j',b'}(w + \delta)),$$

where $2^{\mathcal{A}}$ denotes the powerset of a set $\mathcal{A}$.

**Step 3-1.** First, we claim that there is a sequence $\{\eta'_n\}_n \in \mathcal{E}'$ such that for any $l, i, b$,

$$\mathcal{G}_{l,i,b}(\eta'_n) = \mathcal{G}_{l,i,b}(\eta'_m) \qquad \text{for all } n, m \in \mathbb{N}. \tag{25}$$

Such a sequence always exists as follows: take any $\{\hat{\eta}_n\} \in \mathcal{E}'$ and let $\mathcal{H}_n = (\mathcal{G}_{l,i,b}(\hat{\eta}_n))_{l,i,b} \in \prod_{l,i,b} 2^{\mathcal{I}_{l,i,b}}$; then some element in $\{\mathcal{H}_n\}_n$ must appear infinitely many times in $\mathcal{H}$, because $\prod_{l,i,b} 2^{\mathcal{I}_{l,i,b}}$ is a finite set while $\mathcal{H}$ is indexed over an infinite set; hence, we can choose a desired $\{\eta'_n\}_n$ as a subsequence of $\{\hat{\eta}_n\}_n$. Without loss of generality, we assume that $\{\eta'_n\}_n$ has the minimum value of $\sum_{l,i,b} |\mathcal{G}_{l,i,b}(\eta'_1)|$ over all sequences in $\mathcal{E}'$ satisfying Eq. (25).

**Step 3-2.** Next, we claim that there is a subsequence $\{\eta''_n\}_n$ of $\{\eta'_n\}_n$ satisfying the following: for each $n \in \mathbb{N}$, there exists an open neighborhood $\mathcal{O}''_n$ of $w + \eta''_n$ such that

$$\rho_l(y_{l,j',b'}(\cdot)) = \rho_l(y_{l,j'',b''}(\cdot)) \ \text{ on } \ \mathcal{O}''_n \tag{26}$$

for any $l, i, b$, and $(j', b'), (j'', b'') \in \mathcal{G}_{l,i,b}(\eta''_n) = \mathcal{G}_{l,i,b}(\eta'_1)$. We prove this claim by contradiction.

Suppose that there is no such subsequence. Then, there should exist a subsequence $\{\eta^*_n\}_n$ of $\{\eta'_n\}_n$, some $l^*, i^*, b^*$, and $(j^{*\prime}, b^{*\prime}), (j^{*\prime\prime}, b^{*\prime\prime}) \in \mathcal{G}_{l^*,i^*,b^*}(\eta'_1)$ such that:

$$\rho_{l^*}(y_{l^*,j^{*\prime},b^{*\prime}}(\cdot)) \neq \rho_{l^*}(y_{l^*,j^{*\prime\prime},b^{*\prime\prime}}(\cdot)) \tag{27}$$

on any open neighborhood of $w + \eta^*_n$ for all $n$, and

$$\rho_l(y_{l,j',b'}(w + \eta^*_n)) > \rho_l(y_{l,j'',b''}(w + \eta^*_n)) \tag{28}$$

for all $n, l, i, b$, $(j', b') \in \mathcal{G}_{l,i,b}(\eta'_1)$, and $(j'', b'') \in \mathcal{I}_{l,i,b} \setminus \mathcal{G}_{l,i,b}(\eta'_1)$. We can show this by applying a similar argument used above (to show Eq. (25)), to the fact that $\{(l, i, b, j', b', j'', b'') : l, i, b, \text{ and } (j', b'), (j'', b'') \in \mathcal{G}_{l,i,b}(\eta'_1)\}$ is a finite set. By perturbing $\{\eta^*_n\}_n$ a little bit and using Eqs. (27) and (28), we can construct another sequence $\{\eta^{**}_n\}_n$ such that it still belongs to $\mathcal{E}'$ and

$$\rho_{l^*}(y_{l^*,j^{*\prime},b^{*\prime}}(w + \eta^{**}_n)) \neq \rho_{l^*}(y_{l^*,j^{*\prime\prime},b^{*\prime\prime}}(w + \eta^{**}_n)), \tag{29}$$

$$\rho_l(y_{l,j',b'}(w + \eta^{**}_n)) > \rho_l(y_{l,j'',b''}(w + \eta^{**}_n)), \tag{30}$$

for all $n, l, i, b$, $(j', b') \in \mathcal{G}_{l,i,b}(\eta'_1)$, and $(j'', b'') \in \mathcal{I}_{l,i,b} \setminus \mathcal{G}_{l,i,b}(\eta'_1)$. Further, by applying the same argument used to show Eq. (25), we can take a subsequence $\{\eta^{***}_n\}_n$ of $\{\eta^{**}_n\}_n$ so that it satisfies Eq. (25). Then, one can observe that $\{\eta^{***}_n\}_n \in \mathcal{E}'$ and

$$\sum_{l,i,b} |\mathcal{G}_{l,i,b}(\eta^{***}_1)| < \sum_{l,i,b} |\mathcal{G}_{l,i,b}(\eta'_1)|$$

because $|\mathcal{G}_{l^*,i^*,b^*}(\eta^{***}_1)| < |\mathcal{G}_{l^*,i^*,b^*}(\eta'_1)|$ (by Eq. (29)) and $|\mathcal{G}_{l,i,b}(\eta^{***}_1)| \leq |\mathcal{G}_{l,i,b}(\eta'_1)|$ for all $l, i, b$ (by Eq. (30)). This contradicts to our assumption: $\{\eta'_n\}_n$ has the minimum value of $\sum_{l,i,b} |\mathcal{G}_{l,i,b}(\eta'_1)|$ over all sequences in $\mathcal{E}'$ satisfying Eq. (25). Hence, there should exist a subsequence $\{\eta''_n\}_n$ of $\{\eta'_n\}_n$ satisfying Eq. (26).

**Step 3-3.** Lastly, we show the desired conclusion based on the sequence $\{\eta_n''\}_n$, which satisfies *(i)–(iii)* and Eq. (26). To do so, we claim that for each $n \in \mathbb{N}$, there exists a network $\widetilde{\Psi}_n$ with only differentiable maxpools such that $\widetilde{\Psi}_n = \Psi$ and $D^{\text{AD}}\widetilde{\Psi}_n = D^{\text{AD}}\Psi$ both on some open neighborhood of $w + \eta_n''$. Such a network $\widetilde{\Psi}_n$ exists because the fact that $(\Psi, \mathcal{O}_n'')$ satisfies Eq. (26) implies that we can replace all the non-trivial maxpools in $\Psi$ by differentiable maxpools without changing the value of $\Psi$ and $D^{\text{AD}}\Psi$ on $\mathcal{O}_n''$; here, we use the fact that if $f_1, \ldots, f_p, g_1, \ldots, g_q : \mathcal{A} \to \mathbb{R}$ satisfy $f_i(\cdot) = f_{i'}(\cdot) > g_j(\cdot)$ on $\mathcal{A}$ for all $i, i', j$, then $\max_{p+q}(f_1(\cdot), \ldots, g_q(\cdot)) = \max_1(f_i(\cdot))$ on $\mathcal{A}$ for any $i$ (and $\max_1$ is a differentiable maxpool).

Given the existence of $\widetilde{\Psi}_n$, we can show the following:

$$D\widetilde{\Psi}_n(w + \eta_n'') = D^{\text{AD}}\widetilde{\Psi}_n(w + \eta_n'') = D^{\text{AD}}\Psi(w + \eta_n'') \to D^{\text{AD}}\Psi(w) \qquad \text{as } n \to \infty. \tag{31}$$

The first equality is by *(ii)* and Lemma 13 applied to $\widetilde{\Psi}_n$, where the lemma is applicable because $\widetilde{\Psi}_n$ uses only differentiable maxpools; the second equality is by $D^{\text{AD}}\widetilde{\Psi}_n = D^{\text{AD}}\Psi$ on some neighborhood of $w + \eta_n''$. The convergence in the above equation follows from the definition of $D^{\text{AD}}\Psi$ (see Appendix B), *(i)–(ii)*, the definition of $\mathcal{G}_{l,i,b}$, and two observations: for all $n, l, i, b$,

$$D^{s_{l,j',b'}}\rho_l(y_{l,j',b'}(w)) = D^{\text{AD}}\rho_l(y_{l,j',b'}(w)) \qquad \text{for all } (j', b') \in [M_l] \times [B],$$

$$\frac{\partial^{\text{AD}}\rho_l(y_{l,j',b'}(w))}{\partial w_{1:l}} = \frac{\partial^{\text{AD}}\rho_l(y_{l,j'',b''}(w))}{\partial w_{1:l}} \qquad \begin{array}{l}\text{for all } (j', b') \in \mathcal{G}_{l,i,b}(\eta_n'') \text{ and}\\ (j'', b'') = (\mu_{l,i,b}, \nu_{l,i,b}),\end{array}$$

where the first equality is by the definition of $s_{l,j',b'}$ (see Step 1), and the second equality is by $\mathcal{G}_{l,i,b}(\eta_n'') \subseteq \mathcal{T}_{l,i,b}$ (which follows from *(iii)*), $(\mu_{l,i,b}, \nu_{l,i,b}) \in \mathcal{T}_{l,i,b}$, and the definition of $\mathcal{T}_{l,i,b}$ (see Line 14 of Algorithm 1). From Eq. (31) and the fact that $\Psi = \widetilde{\Psi}_n$ on some open neighborhood of $w + \eta_n''$, $\Psi$ is also differentiable on that neighborhood and $D\Psi(w + \eta_n'') = D\widetilde{\Psi}_n(w + \eta_n'') \to D^{\text{AD}}\Psi(w)$ as $n \to \infty$. From this and *(i)*, we obtain $D^{\text{AD}}\Psi(w) \in \partial^{\text{B}}\Psi(w)$ and this completes the proof of Theorem 8.

## F   ALGORITHM 1 FOR NETWORKS WITH SHARED BIAS AND ReLU

Algorithm 2 is Algorithm 1 for networks with shared bias parameters, where all non-differentiable $\rho_l$ are ReLU with $D^{\text{AD}}\text{ReLU}(0) = 0$.

---

**Algorithm 2** Construction of $\mathcal{P}_l$ for networks with shared bias parameters, where $\rho_l = \text{ReLU}$ if $\rho_l$ is non-differentiable with $D^{\text{AD}}\text{ReLU}(0) = 0$

---

1: **Input:** $l$: the index of the target layer, $z_{0:l-1}$: the outputs of the $0, \ldots, (l-1)$-th layer, $w_{1:l}$: the parameters of the $1, \ldots, l$-th layers, $\{\mathcal{I}_j\}_{j \in [N_l] \times [B]}$: the index sets used by the maxpool function $\phi_l$ at the $l$-th layer.
2: **Initialize:** $\mathcal{P}_l \leftarrow \mathbb{R}^{W_l}$
3: **for** $i \in [M_l] \times [B]$ **do**
4:     $y_i \leftarrow \tau_{l,i}(z_{0:l-1}, w_l)$, $\alpha_i \leftarrow \rho_l(y_i)$, $\pi_i \leftarrow \partial y_i / \partial w_l$
5: **end for**
6: **for** $j \in [N_l] \times [B]$ **do**
7:     $\mathcal{S}_j = \text{argmax}_{i \in \mathcal{I}_j} \alpha_i$
8:     **if** $|\mathcal{S}_j| \geq 2$ and $\max_{i \in \mathcal{I}_j} \alpha_i > 0$ **then**
9:         $i^* \leftarrow$ the index in $\mathcal{I}_j$ that AD uses when computing $D^{\text{AD}}\max(\{\alpha_i\}_{i \in \mathcal{I}_j})$
10:        $\mathcal{T}_j \leftarrow \{i \in \mathcal{S}_j : \partial^{\text{AD}}\alpha_{i^*} / \partial w_{1:l} = \partial^{\text{AD}}\alpha_i / \partial w_{1:l}\}$
11:        $\mathcal{P}_l \leftarrow \mathcal{P}_l \cap \{x \in \mathbb{R}^{W_l} : \langle \pi_{i^*}, x \rangle > \langle \pi_i, x \rangle \text{ for all } i \in \mathcal{S}_j \setminus \mathcal{T}_j\}$
12:    **end if**
13: **end for**
14: **return** $\mathcal{P}_l$

---

## G   EXAMPLES OF IDENTICAL IMAGE PATCHES

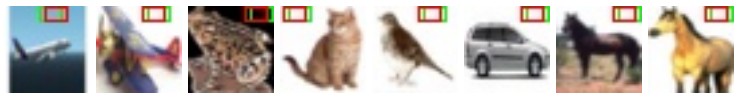

Figure 1: Examples of identical input patches that incur ties in $\max_n$ operations.

In Figure 1, we show example images where a tie occurs in a $\max_n$ operation due to the same image patches that correspond to the receptive fields of the $\max_n$'s inputs. As shown in the figure, due to the nature of the datasets, adjacent image patches are often identical to each other.

## H  DISCUSSION ON FAILURE CASES OF AD AND OUR RESULTS

In this section, we consider three well-known failure cases of AD and discuss how they are handled in Theorems 1, 4–6, and 8. Let $g = h = \mathrm{ReLU}$ with $D^{\mathrm{AD}}g(0) = 0$ and $D^{\mathrm{AD}}h(0) = 1$. Then, AD returns an incorrect output (i.e., a value not in the Clarke subdifferential) in the following cases, even though it uses proxy gradients that are in the Bouligand subdifferential.

- For $f_1(w) = g(w) - g(-w)$, we have $D^{\mathrm{AD}}f_1(0) = 0 \notin \{1\} = \partial^{\mathsf{C}}f_1(0)$.
- For $f_2(w) = g(w) - h(w)$, we have $D^{\mathrm{AD}}f_2(0) = -1 \notin \{0\} = \partial^{\mathsf{C}}f_2(0)$.
- For $f_3(w) = h(-h(w))$, we have $D^{\mathrm{AD}}f_3(0) = -1 \notin \{0\} = \partial^{\mathsf{C}}f_3(0)$.

The three cases are handled in our theorems as follows. First, Theorems 1 and 4 require that a given neural network should have distinct bias parameters. However, none of $f_i$ can be represented by a network with distinct bias parameters, so they simply do not satisfy the conditions in Theorems 1 and 4. Second, Theorem 5 requires that a given network should have fully-connected layers. However, none of $f_i$ can be represented by a network with fully-connected layers, so they do not satisfy the condition in Theorem 5 as well. Third, Theorem 6 requires that a given network should have shared bias parameters and use the same $\lambda$ for all activation functions at the same layer (see the statement of the theorem for details). However, $f_1$ and $f_3$ cannot be represented by a network with shared bias parameters, since the two $g$'s in $f_1$ take arguments of different signs and the two $h$'s in $f_3$ use only one parameter $w$; further, $f_2$ cannot be represented by a network that satisfies the aforementioned condition on $\lambda$, since $g$ and $h$ use the left- and right-hand derivatives at 0, respectively. Hence, none of $f_i$ satisfies the condition in Theorem 6. Finally, none of $f_i$ satisfies the sufficient condition in Theorem 8 (i.e., $\mathcal{P}_l \neq \emptyset$ for all $l$), where each $f_i$ is considered a network with two layers: we have $\mathcal{P}_1 = \emptyset$ for $f_1$ and $f_2$, and $\mathcal{P}_2 = \emptyset$ for $f_3$.

## I  DISCUSSION ON THEOREMS 1 AND 6

In this section, we compare Theorems 1 and 6 with a closely related prior result: Theorem 3.6 in (Lee et al., 2023).

Our Theorem 1 is an extension of Theorem 3.6 in (Lee et al., 2023) in two ways. First, our theorem considers a larger class of neural networks than the previous theorem: the latter restricts a network to have no residual connections (page 9 of (Lee et al., 2023)), while the former does not have this restriction (Section 2.2). Second, for the same network, our Theorem 1 proves the same conclusion given in Theorem 3.6 of (Lee et al., 2023), but under a weaker assumption. The latter theorem states that if proxy gradients are in the Bouligand subdifferential, then AD computes an element of the Clarke subdifferential. Our Theorem 1 extends this result as follows: the same conclusion holds even if we use a wider class of proxy gradients (namely those in the Clarke subdifferential). We believe this extension is an important addition to a line of recent works (e.g., Bertoin et al. (2021); Boursier et al. (2022)) on understanding the effects of the choice of proxy gradients.

Our Theorem 6 further generalizes Theorem 3.6 in (Lee et al., 2023) by considering an even larger class of neural networks. The latter theorem considers networks that do not contain usual residual connections, convolutional layers, and normalization layers such as BatchNorm (pages 4 and 9 of (Lee et al., 2023)) and do not allow minibatched inputs; hence, this prior result is not applicable to most convolutional networks used in practice. In contrast, our Theorem 6 allows a minibatch setup and a network with general residual connections, convolutional layers, and normalization layers (Section 2.2); thus, this result is applicable to a wider range of neural networks including practically-used convolutional networks.

## J   DISCUSSION ON ALGORITHM 1 AND THEOREM 8

In this section, we discuss the computational complexity of Algorithm 1 and provide a comparison between Theorem 8 and prior works.

### J.1   COMPUTATIONAL COMPLEXITY OF ALGORITHM 1

We analyze the computational complexity of Algorithm 1 for the $l$-th layer as follows. To simplify the notation, we assume that memory read/write, addition/multiplication, and computing a proxy gradient take a unit cost. Let $\mathcal{A} \subset [M_l] \times [B]$ be the set of indices $i$ of pointwise activation functions $\rho_l$ whose non-differentiable points are touched (i.e., Line 5 of Algorithm 1 is true). For each index $j \in [N_l] \times [B]$ of maxpool neurons, let $\mathcal{S}_j \subset [M_l] \times [B]$ be the set defined in Line 11 of Algorithm 1, and let $\mathcal{B} \subset [N_l] \times [B]$ be the set of $j$'s that satisfy $|\mathcal{S}_j| \geq 2$ (i.e., Line 12 of Algorithm 1 is true). First, for each element in $\mathcal{A}$, Algorithm 1 adds a constraint to $\mathcal{P}_l$. Next, for each index $j \in \mathcal{B}$, Algorithm 1 computes $\mathcal{T}_j$, which requires at most $|\mathcal{S}_j|$ number of backward passes of AD (up to the $l$-th layer), and then it adds at most $|\mathcal{S}_j| - 1$ constraints to $\mathcal{P}_l$. Finally, Algorithm 1 checks whether $\mathcal{P}_l$ is empty or not, which can be done by solving a linear programming problem (see the last paragraph of Section 3). Hence, the worst-case time complexity of Algorithm 1 can be written as $O(|\mathcal{A}| + (\sum_{j \in \mathcal{B}} |\mathcal{S}_j|) \cdot C_l + D_{W_l, |\mathcal{A}| - |\mathcal{B}| + \sum_{j \in \mathcal{B}} |\mathcal{S}_j|})$, where $C_l$ denotes the cost of a backward pass of AD up to the $l$-th layer and $D_{n,k}$ denotes the cost of solving a linear programming problem with $n$ variables and $k$ constraints. To sum up, Algorithm 1 has a worst-case time complexity that depends on the number of ties arising in maxpools (i.e., $|\mathcal{S}_j|$) and the number of non-differentiability touches in pointwise activation functions (i.e., $|\mathcal{A}|$).

### J.2   EMPIRICAL OVERHEAD OF ALGORITHM 1

For neural networks that have shared bias parameters and use maxpools and ReLUs (with $D^{\text{AD}}\text{ReLU}(0) = 0$) as the only non-differentiable activation functions, Algorithm 1 can be simplified to Algorithm 2 which does not care about whether any input to ReLU touches zero or not (i.e., does not care $\mathcal{A}$ discussed above). We used Algorithm 2 to check $\mathcal{P}_l \neq \emptyset$ in our experiments, and empirically observed that Algorithm 2 incurred not much computational overhead: for training ResNet18 on the CIFAR-10 dataset, the average running times per epoch were 419 seconds with Algorithm 2 and 237 seconds without our algorithms, i.e., additional computational overhead was $\sim 77\%$ of the running time of the vanilla learning algorithm. We further observed that solving linear programming did not incur much overhead ($<1\%$); almost all overhead ($>99\%$) was from computing $\mathcal{S}_j$ and $\mathcal{T}_j$, and this overhead can be significantly reduced if we optimize our naive implementation of Algorithm 2 (e.g., by implementing a native GPU kernel for computing $\mathcal{S}_j$). This relatively small overhead of Algorithm 2 was due to two phenomena we observed (shown in the second and fourth columns of Table 2): ties in maxpools occurred mostly in the first layer, so the backward passes of AD done in Algorithm 2 were very fast; and the number of constraints in $\mathcal{P}_l$ was typically small, so checking the emptiness of $\mathcal{P}_l$ was very fast. To sum up, we observed that the empirical overhead of running Algorithm 2 was relatively low in the training of neural networks.

### J.3   COMPARISON OF THEOREM 8 WITH PRIOR WORKS

Compared to existing results, Theorem 8 has made important contributions in both theoretical and empirical perspectives. Theoretically, Theorem 8 is a strict generalization of Theorem 4.7 in (Lee et al., 2023), one of best known sufficient conditions for AD to compute a Clarke subderivative. More precisely, Theorem 8 not only includes the previous theorem as a special case, but also covers many more cases such as convolutional networks with residual connections and normalization layers (which cannot be covered by the previous theorem). To our knowledge, Theorem 8 is the first sufficient condition that is applicable to practical neural networks. Empirically, Theorem 8 enables us to verify that AD actually computed a Clarke subderivative in several practical learning scenarios (Section 4). To our knowledge, there has been no prior work that empirically verified (or theoretically proved) that AD always outputs a Clarke subderivative in certain learning scenarios; our work is the first such work.

