# OpenReview forum: "What does automatic differentiation compute for neural networks?"
_ICLR.cc/2024/Conference — ICLR 2024 spotlight_

### Official Review · Reviewer_84GA · 2023-10-13

**Soundness:** 3 good
**Presentation:** 2 fair
**Contribution:** 3 good
**Rating:** 6
**Confidence:** 3

**Summary:**

### Edit after rebutal:
I updated my score and my assessement of the paper after reading authors response.

This is an important topic which oughts to be discussed. The authors claim original results on correctness of automatic differentiation in a nonsmooth context based on the existence of bias parameters for each layer.

**Strengths:**

The results are among the only positive ones available in the litterature regarding soundness of AD for nonsmooth neural networks.

**Weaknesses:**

### Edit after rebutal:
the authors responded in a satisfactory way to the concerns raised below.

- Theorem 1 is a minor extension of Theorem 3.6 in Lee et al. (2023).
- Lemma 2 and Lemma 3 are complicated formulations of essentially known facts
- Lemma 3 and Theorem 4 are contradictory
- I suspect that Theorem 4 is flase, as well as theorem 6.
- Conditions 1 and 2 are very complicated without an intuitive explaination of what they mean.
- Theorem 8 lacks discussion on the rate of positive outcomes and complexity.

**Questions:**

Lemma 2 and Lemma 3:  the example in Lemma 2 is explicitely mentioned in Kakade and Lee 2018 and the mechanism in Lemma 3 is exaclty the same, namely "incompatibility with addition". A similar comment holds for Lemma 7. Is there anything else beyond these lemmas? Why not refering to the fact that this is a known issue?

Lemma 3 and Theorem 4 are contradictory. D- and D+ are never explicitely defined, so I assume that for a piecewise analytic function it corresponds to the derivative of the function on the left and on the right at a piece change point. In this case, the Bouligand derivative is {D-,D+} and the clarke is its convex hull. So in Theorem 4, choosing all lambda = 0, one is in the setting of Lemma 3. I suspect that Theorem 4 is false for this reason. Is my reasoning correct?

I really have troubles to understand the difference between Condition 1 and Condition 2 with "trivial max-pool". This is very complicated and I believe there should be a qualitative description of what these conditions mean. For example I cannot tell: is one of the condition more general than the other? Similarly: is there is a difference between Theorem 4 and Theorem 6? Lastly, due to the same concern as above, I suspect that Theorem 6 is false.

Theorem 8 is of little use without a discussion on:
- When does the algorithm stop with Pl non empty? Does it occur often? Does it always occur?
- What is the complexity of the algorithm? How much overhead does it represent compared to AD?

---

> ### Author Response · Authors · 2023-11-15
> **Response to Reviewer 84GA**
>
> We appreciate the reviewer for their time and effort to provide valuable comments. We address all comments of the reviewer, and provide pointers to the corresponding updates in the revised paper. All the updates are color coded in the revised version.
>
> **D- and D+ are never explicitly defined.**
>
> We defined $D^-$ and $D^+$ in Section 2.1 of the initial draft; they denote the left- and right-hand derivatives as the reviewer expected. To improve readability, we have added a pointer to these definitions right after Theorem 4 in the revised draft.
>
> **Lemma 3 and Theorem 4 are contradictory.**
>
> Lemma 3 and Theorem 4 (or Theorem 6) do not contradict each other. Theorem 4 shows that AD computes an element of the Clarke subdifferential of a network (satisfying Condition 1 with distinct bias parameters) if the following condition holds: the proxy gradient $D^{AD} \rho_l(x)$ of $\rho_l$ (i.e., the activation function in the $l$-th layer) can be written as $\lambda_l D^-\rho_l(x) + (1-\lambda_l) D^+\rho_l(x)$, where $\lambda_l \in [0,1]$ should be *fixed across all $x \in\text{ndf}(\rho_l)$ and the layer $l$*. That is, the proxy gradient should be a convex combination of the left- and right-hand derivatives, where the same combination weight $\lambda_l$ is used for all inputs $x$ (within the $l$-th layer).
>
> On the other hand, our counterexample in Lemma 3 uses $\rho_l =\text{HardSigmoid}$ with $D^{AD} \rho_l(-3) = D^{AD} \rho_l(3) = 0$. That is, $D^{AD} \rho_l(x)$ uses the left-hand derivative at $x=-3$ and the right-hand derivative at $x=3$. Due to this, there *cannot exist* a combination weight $\lambda_l \in [0,1]$ such that $D^{AD} \rho_l (x) = \lambda_l D^-\rho_l(x) + (1-\lambda_l) D^+\rho_l(x)$ for all $x \in\text{ndf}(\rho_l)$. Hence, this counterexample does not satisfy the condition in Theorem 4, and thus it is not contradictory to Theorem 4. We have clarified this point in the revised version (page 6).
>
> **I suspect that Theorem 4 is false, as well as theorem 6.**
>
> As we explained above, Theorems 4 and 6 are not contradictory to Lemma 3. We formally prove that the two theorems are true, and present their proofs in Appendices D-E.
>
> **Theorem 1 is a minor extension of Theorem 3.6 in Lee et al. (2023).**
>
> We believe our Theorem 1 is more than a minor extension of Theorem 3.6 in (Lee et al., 2023). First, our theorem considers a larger class of neural networks than the previous theorem: the latter restricts a network to have no residual connections (page 9 of (Lee et al., 2023)), while the former does not have this restriction (page 4 of our paper). Second, for the same network, our Theorem 1 proves the same conclusion given in Theorem 3.6 of (Lee et al., 2023), but under a weaker assumption. The latter theorem states that if proxy gradients are in the Bouligand subdifferential, then AD computes an element of the Clarke subdifferential. Our Theorem 1 extends this result as follows: the same conclusion holds even if we use a wider class of proxy gradients (namely those in the Clarke subdifferential). We believe this extension is an important addition to a line of recent works (e.g., [Bertoin+, Boursier+]) on understanding the effects of the choice of proxy gradients. We have clarified this point in the revised version (page 5).
>
> In addition, we would like to emphasize that our Theorem 6 further generalizes Theorem 3.6 in (Lee et al., 2023) by considering an even larger class of neural networks. The latter theorem considers networks that do not contain usual residual connections, convolutional layers, and normalization layers such as BatchNorm (pages 4 and 9 of (Lee et al., 2023)) and do not allow minibatched inputs; hence, this prior result is not applicable to most convolutional networks used in practice. In contrast, our Theorem 6 allows a minibatch setup and a network with general residual connections, convolutional layers, and normalization layers (page 4 of our paper); thus, this result is applicable to a wider range of neural networks including practically-used convolutional networks.
>
> [Bertoin+] Numerical influence of ReLU'(0) on backpropagation, NeurIPS, 2021.\
> [Boursier+] Gradient flow dynamics of shallow ReLU networks for square loss and orthogonal inputs, NeurIPS, 2022.

---

> ### Author Response · Authors · 2023-11-15
> **Response to Reviewer 84GA**
>
> **Lemmas 2 and 3 are complicated formulations of essentially known facts. Why not referring to the fact that this is a known issue?**
>
> We believe Lemmas 2 and 3 are not just a mere reformulation of known facts. As the reviewer pointed out, we indeed proved these two lemmas motivated by the known fact: the Clarke subdifferential is generally “incompatible with addition”, e.g., as in the example given by Kakade and Lee (2018). However, this incompatibility does not always guarantee the incorrectness of AD. For example, we showed that AD always outputs an element of the Clarke subdifferential under some conditions (Theorems 1, 4, 5, 6, and 8). Given this, what we essentially proved in Lemmas 2 and 3 is the following: under certain conditions, the incompatibility issue can actually show up in some neural networks. In this regard, Lemmas 2 and 3 are not an immediate consequence of the known incompatibility issues.
>
> Following the reviewer’s suggestion, we have added the following clarification to the revised version (page 6): “Lemmas 2 and 3 are based on the incompatibility of the Clarke subdifferential with addition (Clarke et al., 1998; Kakada and Lee, 2018).”
>
> **Conditions 1 and 2 are very complicated without an intuitive explanation of what they mean.**
>
> Condition 1 states that (i) each analytic pre-activation function $\tau_l$ should have no bias parameters at all, or have bias parameters that should not be “shared” across multiple neurons in the same layer (e.g., as in fully-connected layers), and (ii) each activation function $\rho_l$ should apply the same piecewise-analytic function elementwise. Similarly, Condition 2 states that (i) each analytic pre-activation function $\tau_l$ should have no bias parameters at all, or have bias parameters that can be shared across multiple neurons in the same layer (e.g., as in convolutional layers), and (ii) each activation function $\sigma_l$ should be the composition of a maxpool function and a elementwise piecewise-analytic function. To improve readability, we have added this explanation to the revised version (page 4).
>
> **Any difference between Condition 1 and Condition 2 with trivial maxpools? Any difference between Theorem 4 and Theorem 6?**
>
> Condition 2 with only trivial maxpools is a strict generalization of Condition 1: (i) the former includes the latter as a special case, and (ii) the former allows shared bias parameters while the latter allows only distinct bias parameters. (i) can be observed by setting $M_l=N_l=C_l$ and $A_{l,c}=e_c{\bf 1}_B^\top$. (ii) follow directly from the definition of the two conditions. We stated these observations before and after Condition 2 in the initial draft. We have made this difference clearer in the revised draft (page 4).
>
> Since Condition 2 with only trivial maxpools is a strict generalization of Condition 1, Theorem 6 is a strict generalization of Theorem 4 that allows shared bias parameters (which are commonly used in convolutional layers and normalization layers).

---

> > ### Comment · Reviewer_84GA · 2023-11-15
> > **Response to authors rebutal**
> >
> > Let me thank the authors for their detailed answer. I must say that I was very suspicious about the claims that the authors made in their initial draft and the main reason is that I could not understand the mechanisms which would make the chain rule valid. This is due to my own fast reading and the heaviness of the notations, conditions etc ...
> >
> > I would like to revise my jugement about the content of the paper, I believe that the results contained in the paper represent considerable improvements of the state of the art and they do look reasonable, although I did not read all proof details. The addition made to the paper, in particuliar the initial table provides nice clarification.
> >
> > I still believe that the notations are very heavy and the paper would benefit from examples:
> >
> > Well known cases of failure of AD are the following at x = 0:
> > - x -> relu(x) - relu(-x)
> > - x -> relu(x) - relu(x)
> > - x -> relu(-relu(x))
> > - maybe other ones
> >
> > The paper would be much more accessible with a qualitative discussion following condition 1 and 2 to describe how such examples are ruled out by the condition, or after Theorems 4, 5, 6 to describe how the chain rule failure is handled on these examples. Also provide an intuition why algorithms 1 and 2 allow to handle chain rule failure after theorem 8, since relu without bias parameter is a maxpool.
> >
> > I still believe that Theorem 1 is a relatively minor extension of Theorem 3.6 of Lee et al: skiped connection could be modeled by appropriate choice of linear maps and enlarged layer sizes, and if the chain rule works for Bouligand, it should also work for Clarke, the main contribution of Lee et al. being the identification of the systematic presence of bias parameters as a sufficient condition to bypass failures of the chain rule.

---

> > > ### Author Response · Authors · 2023-11-15
> > > **Response to Additional Comments**
> > >
> > > We appreciate the reviewer for re-evaluating our paper (based on our response) and providing additional comments. By the end of the discussion period, we will revise our paper to incorporate your comments, and leave another comment here after uploading an updated version.

---

> > > > ### Author Response · Authors · 2023-11-22
> > > > **Follow-up Response**
> > > >
> > > > We have revised our paper according to the reviewer's comments and uploaded the updated version. High-level changes made in this revision are as follows.
> > > >
> > > > - Added the discussion on the complexity of Algorithms 1-2 and on Theorem 8, to Appendix I (page 24).
> > > > - Added the discussion on well-known failure cases of AD and how they interact with our theorems, to Appendix H (page 25).
> > > > - Added the discussion on Theorems 1 and 6, to Appendix J (page 23).
> > > > - Added a pointer to these discussions, to Section 3 (page 5).

---

> ### Author Response · Authors · 2023-11-15
> **Response to Reviewer 84GA**
>
> **Theorem 8 lacks discussion on the rate of positive outcomes and complexity.**
>
> **(1) When does the algorithm stop with non-empty $\mathcal P_l$?**
>
> One condition that guarantees $\mathcal P_l$ to be non-empty is the condition in Theorem 6: when a network satisfies Condition 2 with shared bias parameters and only trivial maxpools. For networks without shared bias parameters or with non-trivial maxpools, we could not (and we expect to be challenging to) find a generic sufficient condition that guarantees the non-emptiness of $\mathcal P_l$, which in turn implies the correctness of AD. This is why we proposed Theorem 8 (with Algorithm 1) in the first place, and why we empirically checked $\mathcal P_l \neq \emptyset$ using experiments.
>
> **(2) Does it occur often? Does it always occur?**
>
> As stated in Section 4 of the initial draft, we empirically observed that $\mathcal P_l$ was always non-empty during the training of the following networks: fully-connected networks with various activation functions (ReLU6, HardTanh, HardSigmoid) on the MNIST dataset, and various convolutional networks (ResNet18, VGG11, VGG11-BN) on the CIFAR-10 dataset. Together with Theorem 8, this implies that AD always computed an element of the Clarke subdifferential in our experiments.

---

> ### Author Response · Authors · 2023-11-15
> **Response to Reviewer 84GA**
>
> **(3) What is the complexity of the algorithm? How much overhead does it represent compared to AD?**
>
> **Theoretical complexity.** We analyze the computational complexity of Algorithms 1 for the $l$-th layer as follows. To simplify the notation, assume that memory read/write, addition/multiplication, and computing a proxy gradient take a unit cost. Let $\mathcal A \subset [M_l] \times [B]$ be the set of indices $i$ of pointwise activation functions $\rho_l$ whose non-differentiable points are touched (i.e., Line 5 of Algorithm 1 is true). For each index $j \in [N_l] \times [B]$ of maxpool neurons, let $\mathcal S_j \subset [M_l] \times [B]$ be the set defined in Line 11 of Algorithm 1, and let $\mathcal B \subset [N_l] \times [B]$ be the set of $j$’s that satisfy $|\mathcal S_j| \geq 2$ (i.e., Line 12 of Algorithm 1 is true). First, for each element in $\mathcal A$, Algorithm 1 adds a constraint to $\mathcal P_l$. Next, for each index $j \in \mathcal B$, Algorithm 1 computes $\mathcal T_j$, which requires at most $|\mathcal S_j|$ number of backward passes of AD (up to the $l$-th layer), and then it adds at most $|\mathcal S_j|-1$ constraints to $\mathcal P_l$. Finally, Algorithm 1 checks whether $\mathcal P_l$ is empty or not, which can be done by solving a linear programming problem (see the last paragraph of Section 3). Hence, the worst-case time complexity of Algorithm 1 can be written as $O(|\mathcal A|+(\sum_{j\in\mathcal B}|\mathcal S_j|) \cdot C_l+D_{W_l,|\mathcal A| - |\mathcal B| + \sum_{j\in\mathcal B} |\mathcal S_j|})$, where $C_l$ denotes the cost of a backward pass of AD up to the $l$-th layer and $D_{n,k}$ denotes the cost of solving a linear programming problem with $n$ variables and $k$ constraints.
>
> **Empirical overhead.** For neural networks that have shared bias parameters and use maxpools and ReLUs (with $D^{AD}\text{ReLU}(0)=0$) as the only non-differentiable activation functions, Algorithm 1 can be simplified to Algorithm 2 which does not care about whether any input to ReLU touches zero or not (i.e., does not care $\mathcal A$ discussed above; see the second paragraph on page 9). We used Algorithm 2 to check $\mathcal P_l \neq \emptyset$ in our experiments, and empirically observed that Algorithm 2 incurred not much computational overhead: for training ResNet18 on the CIFAR-10 dataset, the average running times per epoch were 419s with Algorithm 2 and 237s without our algorithms, i.e., additional computational overhead was ~77% of the running time of the vanilla learning algorithm. We further observed that solving linear programming did not incur much overhead (<1%); almost all overhead (>99%) was from computing $\mathcal S_j$ and $\mathcal T_j$, and this overhead can be significantly reduced if we optimize our naive implementation of Algorithm 2 (e.g., by implementing a native GPU kernel for computing $\mathcal S_j$). This relatively small overhead of Algorithm 2 was due to two phenomena we observed (shown in the second and fourth columns of Table 2): ties in maxpools occurred mostly in the first layer, so the backward passes of AD done in Algorithm 2 were very fast; and the number of constraints in $\mathcal P_l$ was typically small, so checking the emptiness of $\mathcal P_l$ was very fast.
>
> **Summary.** Theoretically, Algorithm 1 (and Algorithm 2) has a worst-case time complexity that depends on the number of ties arising in maxpools (i.e., $|\mathcal S_j|$) and the number of non-differentiability touches in pointwise activation functions (i.e., $|\mathcal A|$). Empirically, we observed that the overhead of running Algorithm 2 was relatively low in the training of neural networks: for ResNet18 on CIFAR-10, it was ~77% of the total training time without running Algorithm 2.

---

> ### Author Response · Authors · 2023-11-15
> **Response to Reviewer 84GA**
>
> **Theorem 8 is of little use without the aforementioned discussion.**
>
> We emphasize that our Theorem 8 has made important contributions in both theoretical and empirical perspectives, and we believe these points would be more important than the above discussion on the complexity. Theoretically, our Theorem 8 is a strict generalization of Theorem 4.7 in [Lee+ 2023], one of best known sufficient conditions for AD to compute a Clarke subderivative. More precisely, our Theorem 8 not only includes the previous theorem as a special case, but also covers many more cases such as convolutional networks with residual connections and normalization layers (which cannot be covered by the previous theorem). To our knowledge, our Theorem 8 is the first sufficient condition that is applicable to practical neural networks. Empirically, our Theorem 8 enables us to verify that AD actually computed a Clarke subderivative in several practical learning scenarios (Section 4). To our knowledge, there has been no prior work that empirically verified (or theoretically proved) that AD always outputs a Clarke subderivative in certain learning scenarios; our work is the first such work.

---

> ### Comment · Reviewer_84GA · 2023-11-15
> **Clarification on Theorem 8 and Algorithms 1 and 2**
>
> Thanks for the clarification, I believe that this discussion on complexity of Algorithm 1, 2 and Theorem 8 should appear in the paper or in its appendix.

---

### Official Review · Reviewer_XRNZ · 2023-10-30

**Soundness:** 2 fair
**Presentation:** 3 good
**Contribution:** 2 fair
**Rating:** 8
**Confidence:** 1

**Summary:**

This article provides an important step in the direction of providing a theoretical backbone to the behemoth topic of neural networks -- in particular, showing Clarke stationarity is the gold standard for this class of optimization problems, and they provide (modest, but appreciable) steps in this direction. The contribution does not appear to be huge, but the importance of the problem itself and the quality if the writing and results (if they are true) is worthy of publication.

Due to a medical emergency, I unfortunately did not have the time to check the mathematical details and proofs of this article. To
account for this lack of availability, I am providing a low-confidence review with my impressions based on the results and impact to the research community.

**Strengths:**

In my opinion, the question studied in this article is one of the
most fundamental and pressing questions to modern-day theoretical
machine learning. I am happy to see work in this direction.

**Weaknesses:**

Of course, this is also a very challenging problem.
Appropriately, this article appears to be a modest step in the
right direction, with nothing particularly groundbreaking.
Nonetheless, the impact of this work would still be quite relevant.

Subdifferential analysis is a very technical and detailed topic,
and I must express my lack of confidence in the validity of the results.
It also appears that the authors have mixed up a Bouligand (directional) subdifferential
and a Mordukhovich (sequential) subdifferential on page 3 of their
article.
Furthermore -- and this not a qualm with the article specifically --
I have doubts about the ability of an ML
conference (where reviewers have essentially two weeks to review 3+
articles) to appropriately verify the correctness of the proofs
in the 10+ pages of the (often un-reviewed or under-reviewed)
appendices. I am hopeful that other referees have sufficient time to
verify the proofs here, because if these results *are* indeed
accurate, it is my opinion that this article would absolutely be
worthy of publication.

**Questions:**

Due to a medical emergency, I unfortunately do not have the time to properly vet this article. I offer my deep apologies.

---

> ### Author Response · Authors · 2023-11-15
> **Response to Reviewer XRNZ**
>
> We thank the reviewer for their positive evaluation and valuable comments. We address all comments of the reviewer as follows.
>
> **It appears that the authors have mixed up a Bouligand (directional) subdifferential and a Mordukhovich (sequential) subdifferential on page 3 of their article.**
>
> We believe our definition of a Bouligand subdifferential (on page 3) is standard in the literature (e.g., see [Cui+, Definition 4.3.1], [Walther+, Definition 3.2], [Burke+, Definition 9], [Stechlinski+, page 275]). A Mordukhovich subdifferential is a different notion of a generalized derivative, which is known to contain a Bouligand subdifferential (e.g., see [Walther+, Proposition 3.8], [Burke+, page 238]). We clarify that our paper does not use a Mordukhovich subdifferential (and does not mention it).
>
> [Cui+] Modern Nonconvex Nondifferentiable Optimization, SIAM, 2021.\
> [Walther+] Characterizing and Testing Subdifferential Regularity in Piecewise Smooth Optimization, SIAM Journal on Optimization, 2019.\
> [Burke+] The Subdifferential of Measurable Composite Max Integrands and Smoothing Approximation, Mathematical Programming, 2019.\
> [Stechlinski+] Generalized Sensitivity Analysis of Nonlinear Programs, SIAM Journal on Optimization, 2018.

---

### Official Review · Reviewer_uFDP · 2023-10-31

**Soundness:** 4 excellent
**Presentation:** 4 excellent
**Contribution:** 3 good
**Rating:** 8
**Confidence:** 3

**Summary:**

The paper studies the important problem of what is computed by automatic differentiation (AD) when the activation functions are not continuously differentiable, which is the most common case in practice with activations like the ReLU or its variants.

---

The main results are several theorems (Thm 1, Thm 4, Thm 5, Thm 6, and Thm 8) that give sufficient conditions on a broad class of neural network architectures and training regimes (batch sizes) to ensure that AD always produces an element of the clarke subdifferential. Besides these sufficient conditions, there are several results showing how tight these results are, by proving that relaxing some of the sufficient conditions will result in AD producing something that is not an element of the clarke subdifferential.

There are two conditions considered to ensure that AD produces a clarke subgradient and both of them are centered on bias parameters. In the fist condition, the bias parameters must be distinct and the activations are applied componentwise. In the second condition, the bias parameters can be shared and more general activations (e.g., maxpooling, maxpool2d) are considered.

The strategy for showing that these conditions guarantee AD computes a clarke subgradient is to construct a sequence of points converging to the current point, on which the neural network is actually differentiable, and showing that the limits of the gradient on this set are converging to what is computed by AD, i.e., directly showing the definition of the Clarke subgradient is satisfied for what is computed by AD.

---

The ultimate goal of the paper is theoretical but it does include some empirical validations of the theoretical claims made. The claims are tested by running sgd on fully connected networks and convolutional networks, both with activations that are not continuously differentiable.

In the fully connected case (they use 2 hidden layers with dimensions 256 and 64 respectively, trained on MNIST with batchsize 128), they check whether or not the sufficient condition to ensure AD computes a clarke subgradient is satisfied, and indeed it always is using three different activations (ReLU6, HardTanh, and HardSigmoid). They also confirm that the points where the activations are not continuously differentiable are seen by the algorithm (0 times for ReLU6, 9.8 times for HardTanh, and 13.8 times for HardSigmoid; all averaged over 5 runs).

In the convolutional case, they use the ordinary ReLU activation but combine with maxpools (they test 3 architectures - VGG11, VGG11-BN (batchnorm), and ResNet18 all on CIFAR10). They observe that their sufficient condition to ensure that AD computes a clarke subgradient is always satisfied here.

---

Small comment: I find the usage of the word safe in the paper to be a bit weird. Whether or not AD computes a clarke subgradient doesn't make it safe or unsafe, even if it does not compute subgradients AD might still converge to a clarke-stationary point (see Bolte-Pauwels 2020). I also found this sentence “These correctness results show that AD computes the standard derivative at most inputs, yet provide no information about what it computes at the remaining inputs” to be incorrect, Bolte-Pauwels 2020 *does* provide information about what is computed at the remaining inputs - an element of the conservative field is computed and this object can still be used to show convergence of sgd-like algorithms.

**Strengths:**

The paper studies an important problem and gives strong theoretical results that apply to a broad class of neural network architectures and training regimes. Despite being theoretical in nature, the results are quite practical since many of the sufficient conditions are checkable in practice, as demonstrated in secion 4 with experiments on some realistic architectures.

**Weaknesses:**

I feel that the empirical validation section is missing something; it's always checked that the sufficient conditions derived in the paper are holding but it's never empiricaly validated that these sufficient conditions are actually sufficient for guaranteeing that AD computes a clarke subgradient. That being said, I still found the experiments are convincing.

**Questions:**

Something that I didn't understand regarding the notion of distinct bias parameters - is it ever an issue that some bias parameters might be equal during training?

---

> ### Author Response · Authors · 2023-11-15
> **Response to Reviewer uFDP**
>
> We appreciate the reviewer for their positive evaluation and thoughtful feedbacks. We address all comments of the reviewer, and provide pointers to the corresponding updates in the revised paper. All the updates are color coded in the revised version.
>
> **I find the usage of the word safe in the paper to be a bit weird. Whether or not AD computes a Clarke subgradient doesn't make it safe or unsafe.**
>
> We thank the reviewer for pointing this out. According to the reviewer’s comment, we have made the following changes in the revised version (pages 2 and 9) to avoid the word “safe”:
>
> > Before revision: “Then, is AD unsafe without distinct bias parameters and the trivial minibatch size?” // “Our results and analyses would contribute to a better understanding and safer use of AD.”
>
> > After revision: “Then, without distinct bias parameters and the trivial minibatch size, when does AD compute an element of the Clarke subdifferential?” // “Our results and analyses would contribute to a better understanding of AD.”
>
> **I also found this sentence to be incorrect: “These correctness results show that AD computes the standard derivative at most inputs, yet provide no information about what it computes at the remaining inputs.”**
>
> We thank the reviewer for pointing this out. We agree that this sentence is too strong so that it makes an incorrect claim. To weaken this sentence and clarify the work of [Bolte+ 20] (and other works), we have made the following changes in the revised version (pages 1 and 2):
>
> > Before revision: “These correctness results show that AD computes the standard derivative at most inputs, yet provide no information about what it computes at the remaining inputs.” // “We remark that some recent works such as (Bolte and Pauwels, 2020a;b; Lee et al., 2020; Huot et al., 2023) proved the correctness of AD with respect to fundamentally new notions of generalized derivatives (e.g., conservative or intensional derivatives).”
>
> > After revision: “These correctness results show that AD computes the standard derivative at most inputs, yet often provide little information about what it computes at the remaining inputs.” // “We remark that some recent works such as (Bolte and Pauwels, 2020a;b; Lee et al., 2020; Huot et al., 2023) proved the correctness of AD over all inputs, with respect to fundamentally new notions of generalized derivatives (e.g., conservative or intensional derivatives).”
>
> **I feel that the empirical validation section is missing something; it's always checked that the sufficient conditions derived in the paper are holding but it's never empirically validated that these sufficient conditions are actually sufficient for guaranteeing that AD computes a clarke subgradient. That being said, I still found the experiments are convincing.**
>
> We are not sure if we correctly understand this comment, but we provide an answer to the following question: “If AD satisfies the sufficient condition given in Theorem 8 (or Theorem 5), then is the output of AD always an element of the Clarke subdifferential?” Our answer to this question is “Yes”, which is stated in Theorem 8 (or Theorem 5). If this question is not what the reviewer intended to ask, please let us know.
>
> We also note that our sufficient condition in Theorem 8 is not necessary in general. Namely, there is an example in which our condition is violated (i.e., $\mathcal P_l = \emptyset$) but AD returns an element of the Bouligand subdifferential. Consider a network $f=g+h-h$, where $g=\text{ReLU}(x)$ with $D^{AD}g(0)=0$ and $h=\text{ReLU}(x)$ with $D^{AD}h(0)=1$. Then, for an input $x=0$, $\mathcal P_l = (-\infty, 0) \cap (0, \infty) = \emptyset$. However, for this input, AD does return an element of the Bouligand subdifferential: $D^{AD} f(0) = 0 \in \partial^B f(0)$. We think developing necessary and sufficient conditions for verifying the correctness of AD is an interesting research direction.

---

> > ### Comment · Reviewer_uFDP · 2023-11-19
> >
> > Thank you for your comprehensive response. Regarding the empirical evaluation, what I meant was that it would be useful to see an *empirical* validation of the statement:
> >
> > "If AD satisfies the sufficient condition given in Theorem 8 (or Theorem 5), then is the output of AD always an element of the Clarke subdifferential?"
> >
> > for some neural network where one can actually compute explicitly the Clarke subdifferential.

---

> > > ### Author Response · Authors · 2023-11-22
> > > **Response to Reviewer uFDP**
> > >
> > > Thank you for clarifying your question. We point out that computing the Clarke subdifferential of a general neural network is known as a *highly challenging problem* by itself. For instance, even a much simplified version of this problem, namely computing two distinct elements of the Clarke subdifferential, is shown to be NP-Hard [Bolte+]. In addition, we are unaware of any numerical method that can compute (or characterize) the Clarke subdifferential of a neural network. For these reasons, it would be very difficult, if not impossible, to do the empirical validation you suggested, as this requires computing the Clarke subdifferential itself.
> > >
> > > More importantly, we emphasize that Theorem 8 is formally proven; that is, the sufficient condition in the theorem always implies the correctness of AD (with respect to the Clarke subdifferential) for all networks satisfying Condition 2. We also note that Condition 2 is satisfied by all the networks used in our experiments. Hence, we believe that it is not necessary to do any empirical validation of our sufficient condition—and this is one particular advantage of proving something mathematically.
> > >
> > > To help your understanding of Theorem 8, we discuss the theorem and the correctness of AD (with respect to the Clarke subdifferential) using concrete examples whose Clarke subdifferentials are easily computable. Consider a network $\Psi(w) = \text{ReLU}(w) - \text{ReLU}(-w)$, where we use $D^{AD} \text{ReLU}(0)=0$. Then, its Clarke subdifferential at 0 does not include the output of AD at 0: $\partial^C \Psi(0) = \\{1\\}$ and $D^{AD} \Psi(0) = 0 + 0 = 0$. As Theorem 8 states, our sufficient condition in Theorem 8 is indeed violated in this case: $\mathcal P = (-\infty,0) \cap (0, \infty) = \emptyset$. Now, consider a different network $\Psi’(w) = \text{ReLU}(w) - \text{ReLU}(2w)$, where we use the same $D^{AD} \text{ReLU}$. Then, our sufficient condition is satisfied: $\mathcal P = (-\infty,0) \cap (-\infty,0) \neq \emptyset$. As Theorem 8 states, the Clarke subdifferential at 0 indeed includes the output of AD at 0 in this case: $\partial^C \Psi’(0) = [-1, 0]$ and $D^{AD} \Psi’(0) = 0 - 0 \cdot 2 = 0$.
> > >
> > > [Bolte+] On the complexity of nonsmooth automatic differentiation, ICLR, 2023.

---

> ### Author Response · Authors · 2023-11-15
> **Response to Reviewer uFDP**
>
> **Something that I didn't understand regarding the notion of distinct bias parameters - is it ever an issue that some bias parameters might be equal during training?**
>
> We thank the reviewer for raising this question. In our paper, “distinct” bias parameters essentially means that the bias parameters used in different neurons should be all distinct, as variables not as values. It is fine for these bias parameters to take the same values as long as they are distinct as variables.
>
> For instance, consider two pre-activation functions $\tau : \mathbb{R}^2 \times \mathbb{R}^1 \to \mathbb{R}^2$ and $\tau’ : \mathbb{R}^2 \times \mathbb{R}^2 \to \mathbb{R}^2$ defined by $\tau(x_1, x_2, b_1) = (x_1+b_1, x_2+b_1)$ and $\tau’(x_1, x_2, b_1, b_2) = (x_1+b_1, x_2+b_2)$. Then, $\tau$ does not have distinct bias parameters since the two outputs of $\tau$ use the same bias parameter $b_1$. In contrast, $\tau’$ has distinct bias parameters since all of its outputs use distinct bias parameters. Here, the values of $b_1$ and $b_2$ are not important and they do not change the fact that $\tau’$ has distinct bias parameters.

---

### Official Review · Reviewer_DHhS · 2023-11-01

**Soundness:** 4 excellent
**Presentation:** 3 good
**Contribution:** 3 good
**Rating:** 8
**Confidence:** 3

**Summary:**

This paper extends the previous results on the characteristics of auto-differentiation (AD) in modern neural networks. Existing methods were limited in cases where the ADs of individual operations are limited to Bouligand subdifferentials, which is extended to Clarke subdifferentials in this paper. The paper shows that if biases are distinct and the batch size is one, then the overall gradient of a fully connected network using AD is always correct (is an instance of Clarke subdifferentials). If the batch size is larger than one, ADs of individual operations should be a convex combination of left-side and right-side derivatives to have a correct gradient. If there are no distinct biases, then having linearly independent features whenever non-differential boundaries are touched. The paper also provides conditions for CNN (presence of max-pool,  shared bias). Experiments demonstrate that the conditions provided in the paper are empirically correct.

**Strengths:**

Even though auto-differentiations are widely used in practice, it is not well-known how it behaves on non-differentiable regions. These are usually ignored in practice. However, since most modern deep learning is usually performed numerically, these regions can sometimes be problematic. In this respect, accurate knowledge regarding the actual behavior can be beneficial.

This paper extends the existing results to wider conditions with more practical settings, i.e., Clarke subdifferentials, general fully-connected networks, and CNN as well. This significantly improves the usability of such knowledge in real situations (such as defining the individual gradient operation so that the overall gradient can be a Clarke subdifferential.)

I did not check the proofs in detail, but they seem correct.

**Weaknesses:**

There are a wide variety of recent network structures and they go beyond the conditions assumed in this paper. However, considering the nature of incremental improvement in this kind of theoretical work, I believe it is a sufficient contribution.

On page 6, it is said that the input and the hidden dimension are typically larger than the batch size. However, this can be somewhat controversial. There are indeed cases where large batch sizes are used (e.g., early self-supervised learning) for several reasons, e.g., training stability, training time/speed, etc.

**Questions:**

On page 6, when batch size is larger than one, a convex combination of the left-side and right-side derivatives is required. Here, the equation seems to suggest that any element that touches the boundary must have the same combination weights. Is this correct?

---

> ### Author Response · Authors · 2023-11-15
> **Response to Reviewer DHhS**
>
> We appreciate the reviewer for their positive evaluation and valuable comments. We address all comments of the reviewer, and provide pointers to the corresponding updates in the revised paper. All the updates are color coded in the revised version.
>
> **There are a wide variety of recent network structures and they go beyond the conditions assumed in this paper. However, considering the nature of incremental improvement in this kind of theoretical work, I believe it is a sufficient contribution.**
>
> We thank the reviewer for appreciating that our paper makes a sufficient contribution. We would like to add that our results already cover a wide range of network architectures, including fully-connected networks, convolutional networks (with residual connections, normalization layers, and maxpools), and transformer-based networks (with attention layers). Nevertheless, our results do not cover some networks, such as recurrent neural networks and the networks that use non-pointwise non-differentiable functions except maxpools (e.g., bilinear interpolations).
>
> **On page 6, it is said that the input and the hidden dimension are typically larger than the batch size. However, this can be somewhat controversial. There are indeed cases where large batch sizes are used for several reasons.**
>
> We thank the reviewer for pointing this out. According to the reviewer’s comment, we have made the following change in the revised version (page 6):
>
> > Before revision: “Since the input and hidden dimensions $N_0, \ldots , N_{L−1}$ are typically larger than the minibatch size $B$, this condition can be easily satisfied in practical learning setups, which we empirically demonstrate in Section 4.”
>
> > After revision: “If the input and hidden dimensions $N_0, \ldots, N_{L−1}$ are larger than the minibatch size $B$ (which often occurs in practical learning setups), this condition can be easily satisfied; we empirically demonstrate this in Section 4.”
>
> **On page 6, when batch size is larger than one, a convex combination of the left- and right-side derivatives is required. Here, the equation seems to suggest that any element that touches the boundary must have the same combination weights. Is this correct?**
>
> Yes, you are correct. The combination weight $\lambda_l$ should remain the same for all inputs $x$ within the same layer (say the $l$-th layer). That is, for each layer $l$, there should exists a single constant $\lambda_l$ such that $D^{AD}\rho_l(x) = \lambda_l D^{-}\rho_l(x) + (1-\lambda_l) D^{+}\rho_l(x)$ for all $x$. Note that $\lambda_l$ can be different for different $l$.

---

> > ### Comment · Reviewer_DHhS · 2023-11-21
> > **Thank you for the answers.**
> >
> > Thank you for the answers. I'm also satisfied with the revision. I'll keep my original score.

---

### Official Review · Reviewer_FQtB · 2023-11-02

**Soundness:** 3 good
**Presentation:** 3 good
**Contribution:** 2 fair
**Rating:** 6
**Confidence:** 2

**Summary:**

The paper studies the problem of automatic differentiation of neural networks with non-smooth operations such as ReLu activation and max-pooling. It provides some theoretical results.

**Strengths:**

1. The theoretical results are rich.
2. The numerical results are consistent with the theoretical results to some extent.

**Weaknesses:**

1. The results in the theorems are not intuitive and are difficult to follow.
2. It is not clear how the numerical or theoretical result given by gradient descent differs from that given by subgradient descent.

**Questions:**

1. Are the results in Table 2 given by gradient descent, subgradient descent, or Clarke subgradient descent?
2. Are the theoretical results applicable to other nonsmooth activation functions such as step function?

---

> ### Author Response · Authors · 2023-11-15
> **Response to Reviewer FQtB**
>
> We thank the reviewer for their positive evaluation and valuable comments. We address all comments of the reviewer, and provide pointers to the corresponding updates in the revised paper. All the updates are color coded in the revised version.
>
> **The results in the theorems are not intuitive and are difficult to follow.**
>
> In the initial draft, we gave an informal overview of our results in Section 1 (pages 2-3), and provided a more detailed yet still high-level description of our results right after each result (pages 5-8). For example, Theorem 1 is informally introduced in Section 1: “Theorem 1 shows that AD always computes an element of the Clarke subdifferential, if the minibatch size is one and all non-differentiable neurons have distinct bias parameters.” The theorem is explained in more detail yet still informally in Section 3.1: “Theorem 1 states that if a network $\Psi$ has distinct bias parameters and the minibatch size is one, then AD computes an element of the Clarke (or Bouligand) subdifferential of $\Psi$ as long as the proxy gradient $D^{AD} \rho_l(x)$ is an element of the Clarke (or Bouligand) subdifferential for all $l$ and $x$.” We hope these informal and/or high-level explanations would make readers easier to follow our results.
> In the revised draft, we have further improved readability by adding more explanations on our problem setup (Conditions 1 and 2) and more discussions on our theoretical results (pages 4–6).
>
> **It is not clear how the numerical or theoretical result given by gradient descent differs from that given by subgradient descent.**
>
> We do not compare gradient descent and subgradient descent algorithms. The main objective of our paper is to understand the output of AD when neural networks contain non-differentiable activation functions such as ReLU and maxpools. Our theoretical results aim to characterize sufficient conditions under which AD always computes an element of the Clarke subdifferential (Theorems 1, 4, 5, 6, and 8), and other conditions under which AD might not do so (Lemma 2, 3, and 7). Our numerical results aim to verify whether our sufficient conditions are satisfied or not when we train neural networks via minibatch stochastic gradient descent (SGD), where the “gradients” used in SGD are computed by AD. We empirically observed that our sufficient conditions were always satisfied, i.e., AD always computed an element of the Clarke subdifferential in our experiments.
>
> **Are the results in Table 2 given by gradient descent, subgradient descent, or Clarke subgradient descent?**
>
> As we noted in Section 4, all the results in Table 2 are given by minibatch stochastic gradient descent (SGD) algorithm implemented in PyTorch, where the “gradients” used in SGD are computed by AD (which is standard in practice). Hence, this learning algorithm is in general not a gradient, subgradient, or Clarke subgradient descent algorithm, since AD might not compute the true gradient or a (Clarke) subgradient for general neural networks (e.g., Lemmas 2 and 3). However, in our experiments, we verified that our sufficient conditions for AD to compute an element of the Clarke subdifferential were always satisfied. This result implies that the networks in Table 2 were in fact trained by a Clarke subgradient descent algorithm.
>
> **Are the theoretical results applicable to other nonsmooth activation functions such as step functions?**
>
> All of our results hold for any piecewise-analytic activation function (Definition 1), which includes ReLU, Leaky-ReLU, HardSigmoid, HardTanh, etc. The step function, on the other hand, is not piecewise-analytic since it is discontinuous, so our results do not apply to it.
>
> Regarding step functions, we note that the Clarke subdifferential has been defined and studied only for continuous functions (especially locally Lipschitz functions), and not for discontinuous ones (see, e.g., [Clarke+, Chapter 2.6], [Cui+, Chapter 4.3]). Nevertheless, computing a Clarke subderivative of a locally Lipschitz network that may contain discontinuous activation functions could be an interesting research problem: e.g., $\Psi(x) = {\bf 1}[x>0] + {\bf 1}[x\le0] = 1$ is an example of such networks.
>
> [Clarke+] Optimization and nonsmooth analysis, SIAM, 1990.\
> [Cui+] Modern Nonconvex Nondifferentiable Optimization, SIAM, 2021.

---

### Public Comment · ~Sanghyuk_Chun1 · 2024-05-03
**Official implementation**

We have released the code! Please check the following repository if you are interested in:
https://github.com/SanghyukChun/ad_correctness

Best,

Authors

---

### Meta-Review · Area_Chair_CAUU · 2023-12-05

**Metareview:**

The paper deals with automatic differentiation in neural networks with non-smooth operations. It extends the understanding of AD behavior in non-differentiable regions and provides conditions under which AD yields elements of the Clarke subdifferential. Reviewers generally recognize the paper's theoretical depth and relevance. To enhance its impact, the authors should address the reviewers' concerns, particularly by clarifying the theoretical results.

**Justification For Why Not Higher Score:**

The theoretical contributions, while significant, lack intuitive clarity.

**Justification For Why Not Lower Score:**

The paper presents valuable theoretical insights.

---

### Decision · Program_Chairs · 2024-01-16

Accept (spotlight)